# ZERO-SHOT NATURAL LANGUAGE EXPLANATIONS

**Fawaz Sammani & Nikos Deligiannis**
ETRO Department, Vrije Universiteit Brussel, Pleinlaan 2, B-1050 Brussels, Belgium
imec, Kapeldreef 75, B-3001 Leuven, Belgium
`fawaz.sammani@vub.be, ndeligia@etrovub.be`

## ABSTRACT

Natural Language Explanations (NLEs) interpret the decision-making process of a given model through textual sentences. Current NLEs suffer from a severe limitation; they are unfaithful to the model's actual reasoning process, as a separate textual decoder is explicitly trained to generate those explanations using annotated datasets for a specific task, leading them to reflect what annotators desire. In this work, we take the first step towards generating faithful NLEs for any visual classification model without any training data. Our approach models the relationship between class embeddings from the classifier of the vision model and their corresponding class names via a simple MLP which trains in seconds. After training, we can map any new text to the classifier space and measure its association with the visual features. We conduct experiments on 38 vision models, including both CNNs and Transformers. Our method outperforms supervised baselines on many metrics, while remaining comparable on others. In addition to NLEs, our method offers other advantages such as zero-shot image classification and fine-grained concept discovery, each outperforming baseline methods. Finally, we also show that our method achieves state-of-the-art results on zero-shot image captioning.

## 1 INTRODUCTION

Consider the example in Figure 1, where an image classifier incorrectly identifies the input image as a *volcano* rather than a *leatherback turtle*. Why was this decision made and which features of the image led to this prediction? When we apply popular attribution techniques such as the CAM family (Zhou et al., 2015; Selvaraju et al., 2019; Jiang et al., 2021), the explanations are uninformative; perhaps they indicate that the model focused on the main object, but they fail to provide deeper insight. What specific elements within the attributed region influenced the prediction? What features of this image category were significant? Existing attribution techniques offer high-level interpretations and do not break down the reasoning behind the decision. On the other hand, Natural Language Explanations (NLEs) (Park et al., 2018; Sammani et al., 2022) offer detailed interpretations in a textual format. NLEs not only resolve the limitations of attribution methods but also present explanations in a more human-friendly manner, accessible even to a layman user. As illustrated in Figure 1, the NLE reveals that the classifier mistook the image for a volcano due to the bright red glow, resembling erupting lava. This insight allows us to deduce that the classifier is not robust to color shifts.

However, current NLE models are explicitly trained to generate textual explanations using annotated datasets. This renders NLEs as unfaithful, as they reflect what dataset annotaters desire rather than the model's true reasoning process. Other than that, NLE models are characterized with the *shortcut bias* problem, as shown in Sammani & Deligiannis (2023), rendering the explanation meaningless despite achieving state-of-the-art results on natural language generation metrics.

In this work, we aim to generate faithful NLEs in a zero-shot setting for a visual classifier model. This boils down to the question: *how can we access the visual features via any natural text?* To achieve this, we require an intermediary between visual features of the model, and natural language text. Notably, the discrete labels of the classifier correspond to class names in text format, and the classifier weights of each class serve as a embedding vector representation for that class. These embeddings effectively encode semantics; similar objects are close together in the embedding space. Figure 2 illustrates a visualization of those embeddings from a ResNet-50 model for certain classes, using t-sne (van der Maaten & Hinton, 2008). As shown, ocean fish such as different types of sharks

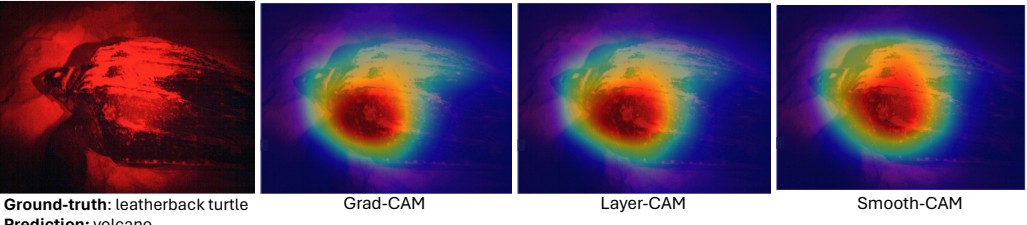

**Ground-truth:** leatherback turtle
**Prediction:** volcano

Grad-CAM      Layer-CAM      Smooth-CAM

**NLE (Ours):** this is a volcano because it shows fire burning on the volcano lava flows, a bright red glow erupts

Figure 1: Comparison between some attribution methods and a Natural Language Explanation (NLE) for explaining the prediction of an image as a *volcano*

and rays form one cluster, while freshwater fish proximally form another cluster. Similarly, small birds such as *robin* and *jay* form a distinct cluster, and so on. Given that discrete labels represent class names, the classification layer of the vision model performs a closed-set visual-textual search over all possible classes, with the predicted class being the one with the highest cosine similarity between the visual features and class embeddings. The classification layer therefore retrieves the class closest to the image. This implies that those class embeddings and the visual features share a common space. Therefore, if we can learn a mapping from the available text data (class names) to the vector embeddings of the classes (classifier weights), we have essentially learned the visual-semantic space of the vision classifier, which then allows us to translate any new text (e.g., a new class, a sentence, or a concept) into that embedding space and measure its association with the visual features.

Once the mapping is learned, we employ an off-the-shelf frozen language model and utilize inference-time prefix-tuning (Li & Liang, 2021) to generate a NLE that maximize the similarity with the visual features. Additionally, the learned mapping offers us with several other applications, such as concept discovery and zero-shot classification.

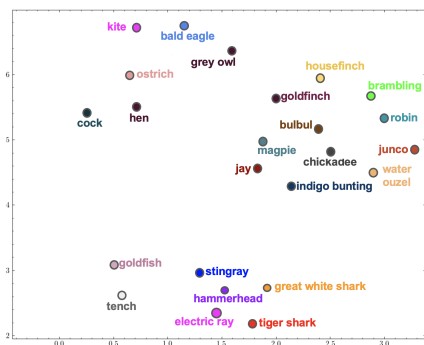

Figure 2: t-sne visualization of a ResNet-50 class weight embeddings

To summarize, our contributions are as follows: 1) We are the first to generate NLEs in a zero-shot manner for visual classifiers that are also faithful to the model, achieving performance that is comparable to, and in many cases surpasses supervised baselines. 2) We demonstrate the effectiveness of our approach in three additional applications: zero-shot concept discovery, image classification and image captioning, showing significant improvements over baselines. 3) We evaluate a wide set of 38 vision classifiers, including both CNNs and Transformers.

## 2 RELATED WORK

**Natural Language Explanations:** Early works on NLEs for vision and vision-language tasks include Hendricks et al. (2016); Park et al. (2018); Marasović et al. (2020); Kayser et al. (2021). The standard pipeline typically consists of a task model (e.g., a classifier) for prediction, coupled with an explainer model (e.g., GPT-2 (Radford et al., 2019)) to generate an explanation for the prediction. NLX-GPT (Sammani et al., 2022) proposed a unified approach, combining both models into a single, compact system that simultaneously generates and explains answers in a single text stream using a single causal language modeling objective, ensuring that the reasoning process for the explanation aligns with that of the prediction. Building on this, Uni-NLX (Sammani & Deligiannis, 2023) further consolidates multiple tasks into one model, achieving the ability to perform seven tasks simultaneously. Multimodal-CoT (Zhang et al., 2023) builds upon the Chain of Thought Prompting technique and instead generates a rationale (explanation) prior to generating an answer, which serves as a reasoning step for inferring the answer. In DeViL (Dani et al., 2023), an autoregressive generator is trained on the conceptual captions image captioning dataset (CC3M) (Sharma et al., 2018) using

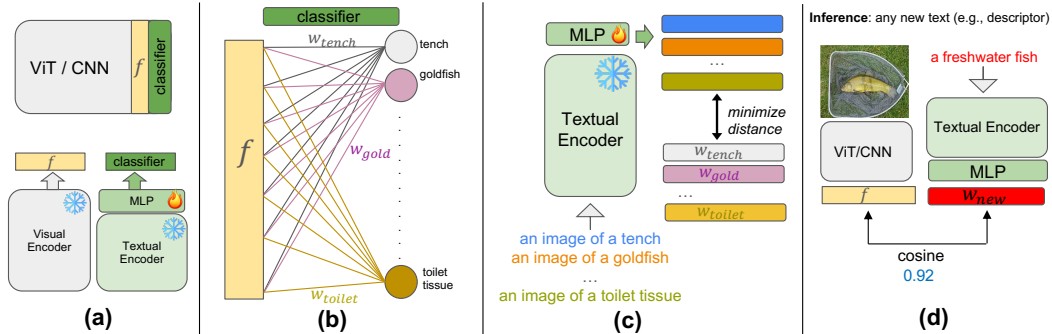

Figure 3: An overview of our method to learn class embeddings. **(a)** We transform the visual classifier into a visual feature encoder and a text encoder paired with an MLP module that outputs classifier weights. **(b)** The classifier weights, which are the class embeddings, serve as ground-truth data to train the MLP module. **(c)** We learn the class embeddings by converting the class name into a text prompt, encoding it with the text encoder and MLP module, aiming to minimize the distance between the MLP output and the ground-truth data. **(d)** Once the MLP module is trained, we can map any new text to the classifier embedding space, which now shares the same space as the visual features, allowing us to measure the association between them. ❄ indicates that the module is frozen, while 🔥 indicates trainable.

a subset of features from different layers implemented by applying Dropout. However, all these methods rely on annotated datasets consisting of (prediction-explanation) pairs. As a result, the explanations generated reflect the reasoning of the annotators rather than that of the model. This is particularly evident in DeVIL, which treats image captions as explanations. On the other hand, our NLEs are generated in a zero-shot manner in a way that maximizes the classifier's visual feature space, ensuring that they are faithful to the classifier's reasoning process.

**Zero-Shot Textual Explanations:** ZS-A2T (Salewski et al., 2023) is the work most closely related to ours, as it translates attention maps into textual explanations in a zero-shot manner. There are also several studies (Menon & Vondrick, 2023; Shtedritski et al., 2023) which offer concept-based explanations (short textual descriptors) rather than full sentence-based explanations for model predictions. However, all of these approaches are constrained to vision-language models that are trained to learn a shared vision-language space through a contrastive objective, meaning they cannot be applied to visual classifiers. In contrast, our NLEs are classifier-agnostic, do not depend on a contrastive training objective, and provide full sentence-based explanations.

## 3 METHOD

**Notations:** Consider a visual classifier model $M$ that we want to interpret, which can be of any architecture (*e.g.,* CNN, Transformer or Hybrid). $M$ consists of a visual feature encoder $M_V$, and a classification layer $\mathbf{W}$. $M_V$ maps an input image $x$ into a feature vector $f$. That is: $M_V(x) = f \in \mathbb{R}^d$. Usually, $f$ is the result of average or max pooling, or is the result of the `[CLS]` token in some Transformer models. The visual feature vector $f$ is then fed to the classifier layer $\mathbf{W} \in \mathbb{R}^{d \times C}$ with $C$ classes. Therefore, the matrix $\mathbf{W}$ consists of all class embedding vectors which encode the semantics of a class: $\mathbf{W} = [\mathbf{w}_{c_1}, \mathbf{w}_{c_2}, \ldots, \mathbf{w}_C]$, and $|W| = C$. The prediction $c_p$ is given by $\arg\max(f \cdot \mathbf{W})$. Unlike the visual features $f$, note that $\mathbf{W}$ is fixed within the network and independent of the input image $x$. We consider an off-the-shelf textual encoder $M_T$ capable of encoding sentences, with the only exception that this textual encoder does not also encode image information (as in the case of the CLIP textual encoder (Radford et al., 2021)). The textual encoder $M_T$ takes as an input a natural text $t_i$ and produces a sentence embedding $s_i$ of $k$ dimensions. That is, $M_T(t_i) = s_i \in \mathbb{R}^k$.

### 3.1 LEARNING CLASS EMBEDDINGS

The visual classifier $M$ is a closed-set classifier. Our goal is to expand it into an open-set visual classifier capable of understanding any natural text. Specifically, we want to access or query the

classifier via text. The simplest approach is to align a text encoder with the visual features of the classifier, and then access those features through text. This is the contrastive language-image pretraining paradigm that CLIP (Radford et al., 2021) employs. However, this method is 1) not faithful to the classifier and 2) poses significant technical challenges. More details are provided in Section M of the appendix. To address these issues, we aim to learn a regression function that maps text to the classifier's embedding space. Note that this is an ill-posed problem, meaning there is either no unique solution or the correct solution is difficult to determine. While we may not be able to learn the exact ground-truth values of the $d$-dimensional class embedding vectors $\mathbf{W}$, we can still capture their underlying semantics.

The classification layer $\mathbf{W}$ contains class embeddings which share the same space as the visual features $f$. Drawing an analogy with CLIP (Radford et al., 2021), $\mathbf{W}$ can be thought of as the output of a text encoder that is trained to be aligned with the visual features $f$ through an alignment loss such as contrastive learning. However, in practice, $M_T$ is a pretrained off-the-shelf text encoder that is not trained to align with $f$. Drawing inspiration from Christensen et al. (2023), we introduce a small learnable module comprising a simple Multi-Layer-Perceptron (MLP), which is trained to map text to the classifier embedding space (Figure 3**a**). We first map every textual class name $c_i$, where $i = 1, 2, \ldots, C$, to a corresponding natural text prompt $t_i$, specified as: *an image of a* $\{c_i\}$. For example, if $c_i$ is the class *tench*, then $t_i$ is: *an image of a tench*. Given that we want to learn a mapping from $t_i$ to the class embedding $\mathbf{w}_{c_i} \in \mathbf{W}$, the only ground-truth data available to us is $\mathbf{W}$, which are the fixed $C$ class embeddings of the classifier (Figure 3**b**). For ImageNet-1k (Deng et al., 2009), $C = 1,000$, presenting a highly scarce data scenario. Nevertheless, as we will elaborate later, this challenge can still be addressed using specific techniques. Each prompt $t_i$ is then encoded with the text encoder $M_T$ to produce a text embedding representation $s_i$, which we feed to a shared **MLP** layer. The regressed class embedding for the class $c_i$ is defined as $\hat{\mathbf{w}}_{c_i} = \mathbf{MLP}(s_i)$. The prediction $c_p$ can then given by $\arg\max(f \cdot \hat{\mathbf{W}})$, where $\hat{\mathbf{W}} = [\hat{\mathbf{w}}_{c_1}, \hat{\mathbf{w}}_{c_2}, \ldots, \hat{\mathbf{w}}_C]$.

The **MLP** is trained to minimize the distance between the regressed class embedding $\hat{\mathbf{w}}_{c_i}$ and the ground-truth embedding $\mathbf{w}_{c_i}$ (Figure 3**c**). Note that the only trainable module is the **MLP**, and the visual classifier $M_V$ and text encoder $M_T$ are kept frozen. We employ the cosine distance instead of other functions like L1 or Mean Squared Error (MSE) loss, as cosine loss incorporates angular information. This approach yields improved performance, as demonstrated in the ablation experiments in Section D of the appendix. The loss objective $L$ is then given by:

$$L = \sum_{i=1}^{C} \left( 1 - \frac{\hat{\mathbf{w}}_{c_i} \cdot \mathbf{w}_{c_i}}{\|\hat{\mathbf{w}}_{c_i}\| \|\mathbf{w}_{c_i}\|} \right) \tag{1}$$

Training the MLP takes roughly 10 seconds on a single moderate GPU, and is therefore considered negligible and can be applied instantly to any visual classifier model. Once the MLP is trained, we can map any new text (*e.g.,* a descriptor, new class, or sentence) to the class embedding space. This mapped text now shares the same embedding space as the visual features $f$, and we can measure the association of that new text to $f$ (Figure 3**d**). It is worth noting, that in coarse or fine-grained classification, some class names (e.g., "tench") may be unfamiliar to a text encoder trained on general language. However, because their corresponding class embedding is close to that of another class that the text encoder understands (e.g., "goldfish") (see Figure 2), the MLP can infer that "tench" is a fish, even though the word "fish" is not part of the class name of "tench". Furthermore, it is important to note that we only use the class name, and no other supplementary information. More details regarding this can be found in Section L of the appendix.

As mentioned, we are faced with an extremely scarce data scenario. We therefore utilize two simple but highly effective data augmentation techniques, both which allow us to create training points which are in proximity to the original training sample.

**Input Dropout**: This follows the conventional Dropout technique (Srivastava et al., 2014). We randomly drop 50% of $s_i$ (the input features to the **MLP**) with a probability of 0.3. The regressed class embedding is then defined as $\hat{\mathbf{w}}_{c_i} = \mathbf{MLP}(\mathbf{drop}(s_i))$. As shown in Section D of the appendix, this simple technique already boosts performance by a large margin.

**Learned Soft Dropout**: We propose to implement a learned dropout mechanism, where instead of completely zeroing out features randomly, we learn how to selectively scale them with values between 0 and 1. We term this approach as *Soft Dropout*. We implement this via a Gated Linear

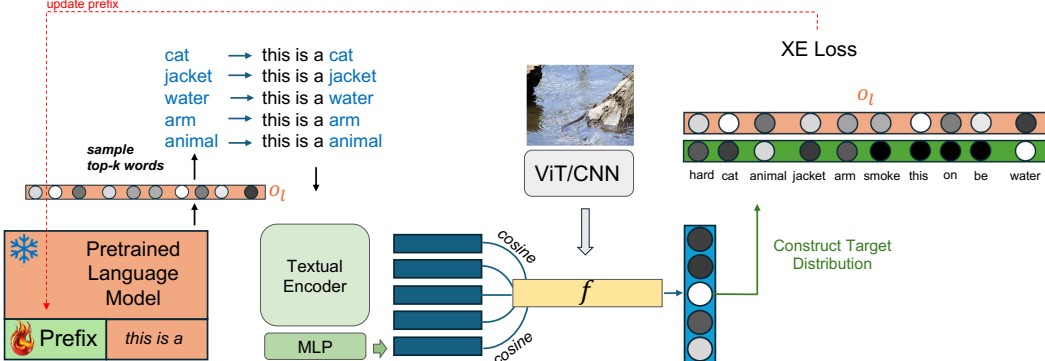

Figure 4: An overview of how we update the prefix. The example is shown for the first timestep $l = 1$ at iteration $i = 1$, with $K = 5$ and a hard prompt set as {*this is a*}. **cosine** indicates cosine similarity.

Unit (GLU) activation function (Dauphin et al., 2016) on the MLP output. The regressed class embedding is then defined as $\hat{\mathbf{w}}_{c_i} = \mathbf{GLU}(\mathbf{MLP}(\mathbf{drop}(s_i)))$. Since GLU is applied independently to *each* dimension of the **MLP** output, it scales different dimensions of the learned class embedding, effectively associating the space around the training point to the corresponding class. As shown in Section D of the appendix, this technique further provides significant improvements in performance in such a scarce data scenario.

## 3.2 GENERATING ZERO-SHOT NLES

Once we establish the mapping function **MLP** which projects the textual features into the same space as the visual features $f$, we can optimize an off-the-shelf pre-trained language model (PLM)—capable of generating coherent text—to generate a NLE that maximizes the similarity with the visual features $f$. We freeze the PLM to keep its powerful generation capability, and instead use prefix-tuning (Li & Liang, 2021), an efficient-finetuning method which attaches learnable vectors in the embedding space of the PLM. We follow an inference-time approach where we optimize learnable prefixes for each sample individually. Unlike supervised models which require a single forward pass at inference, this approach is more time-consuming. However, it allows for much greater flexibility in the generated text, operates in an open-set environment (generating words outside the dataset corpus), and can be adapted to any example or classifier on-the-fly. We build upon the approach introduced by Tewel et al. (2021). An overview of this process is shown in Figure 4. Given a PLM (*e.g.,* GPT-2), we attach randomly initialized learnable prefixes to it to steer the frozen language model to generate explanations of the classifier. Here, the learnable prefixes are set as initial key-value pairs in each attention block of the Transformer, such that each generated word in the explanation can attend to these prefixes. At each timestep $l$, we sample the top-$K$ tokens from the PLM output vocabulary distribution $o_l$, which act as $K$ continuation tokens of the currently generated explanation. These $K$ potential sentences are then passed to the text encoder $M_T$, followed by the learned **MLP** to produce $K$ vectors that share the same embedding space as the visual features $f$. The cosine similarity between each of those vectors and the visual features $f$ is then computed, yielding $K$ similarity scores with the visual feature vector $f$. These scores, normalized with softmax, form a target distribution to train against $o_l$ with the standard Cross-Entropy loss, and the prefixes of the PLM are updated with backpropagation. We then run the PLM again with the updated prefixes and we sample from the output distribution the most likely token. Namely, per timestep, $K$ token are initially produced to update the prefixes but one token is finally sampled (after the prefixes are updated) as the continuation of the explanation. We run the above process for $L$ timesteps. We set $L$ as the maximum sequence length we want for the explanation, or until the $<.>$ token is reached. After these $L$ timesteps, one iteration will be concluded, and one explanation will be generated. Therefore, each iteration generates one complete explanation. The sampled $K$ tokens which have their continuation relevant to the features $f$ will have their scores increased during the generation process, while non-relevant tokens will have their scores decreased. We train the prefixes for $I$ iterations, producing $I$ explanations. Because the similarity between the **MLP** output and the visual features $f$ is used to construct the target distribution for training the prefixes, we choose a different similarity measure—not biased toward the process and metric used to create these explanations—to select the best explanation from

the $I$ generated ones. Specifically, we select the explanation that maximizes the CLIP-Score as the final explanation. More details are provided in Section I of the appendix.

# 4  EXPERIMENTS

In this section, we present both quantitative and qualitative experiments. Implementation details are provided in Section I of the appendix. We evaluate a diverse set of 38 vision models. A selection of these models is included here, with the remainder provided in the appendix. For CNNs, we consider the following family of models (each with several variants): Residual Networks (ResNets) (He et al., 2015), Wide ResNets (Zagoruyko & Komodakis, 2016), ResNeXts (Xie et al., 2016), Densely Connected Networks (DenseNets) (Huang et al., 2016), EfficientNetv2 (Tan & Le, 2021), ShuffleNetv2 (Ma et al., 2018), MobileNetv3 (Howard et al., 2019), ConvNeXts (Liu et al., 2022) and ConvNeXtv2 (Woo et al., 2023). For Transformers, we consider the following family of models (each with several variants): Vision Transformers (ViTs) (Dosovitskiy et al., 2021), Swin Transformer (Liu et al., 2021), BeiT (Bao et al., 2022), DINOv2 (Oquab et al., 2024) and the hybrid Convolution-Vision Transformer CvT (Wu et al., 2021). All models are pretrained on ImageNet-1K (Deng et al., 2009). Models with the subscript *pt* indicate that the model was pretrained on ImageNet-21k before being finetuned on ImageNet-1K. Models with a subscript *v2* are trained with the new recipe from PyTorch (Vryniotis, 2021). Finally, BEiT, DINOv2 and ConvNeXtv2 are pretrained in a self-supervised manner before being finetuned on ImageNet-1k.

## 4.1  LEARNED CLASS EMBEDDINGS

To evaluate the learned class embeddings, we employ zero-shot image classification as a benchmark. Since our approach can map any new text to class embeddings (*i.e.,* class weights), we can treat new unsen classes as input text. This implies that by mapping $n$ new classes to class embeddings, we effectively extend our classifier by $n$ additional classes. It is important to note that we are referring to the classical zero-shot classification paradigm (Lampert et al., 2014; Xian et al., 2017b), unlike CLIP (Radford et al., 2021) which addresses zero-shot *transfer*. As a baseline, we compare against ICIS (Christensen et al., 2023), the current state-of-the-art in zero-shot image classification using text-only training. For a fair comparison, we re-implemented their method, utilizing the same text encoder, dataset splits and class names as in our approach. We use the challenging ImageNet-1K dataset as our benchmark, splitting its $1,000$ classes into $900$ for training and $100$ for testing. For validation and hyperparameter tuning, we use 100 non-overlapping classes from the ImageNet-21K dataset.

We follow the standard evaluation protocol for zero-shot image classification which includes two settings: the *Generalized Zero-Shot Setting* and the *Zero-Shot Setting*. The Generalized Zero-Shot Setting is the most important as it reflects real-world scenarios. In this setting, we generate class weights for both the seen (training) classes, and the unseen (testing) classes. For ImageNet, these are all the $1,000$ classes. The image is then classified into one of these classes by comparing its features $f$ against all these classes, and the class with the highest similarity is predicted. "Train" refers to the results on the training set, while "Test" refers to the results on the testing set. We report the standard Top-1 (@1) and Top-5 (@5) accuracy metrics. In the Zero-Shot Setting, we generate weights for the unseen (testing) classes only, and the image is classified solely among the testing classes.

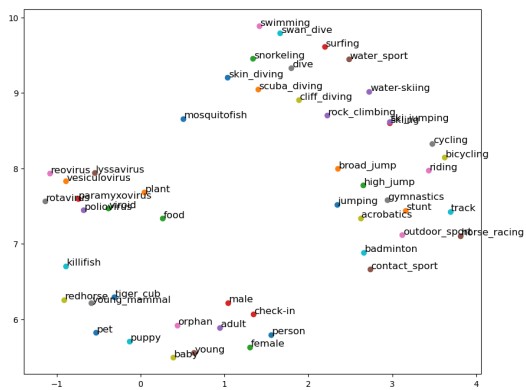

Figure 5: t-sne embeddings on new unseen classes from ImageNet-21K, for a ResNet-50

In the case of ImageNet, the image features $f$ are compared only against the 100 testing classes, and the class with the highest similarity is predicted. However, this setting is less realistic, as in practice we do not know which classes are unknown. Results are presented in Table 1. Our method significantly outperforms ICIS on the testing set in the generalized zero-shot setting and, in most

Table 1: Quantitative Results on evaluating class embeddings via zero-shot image classification on ImageNet-1K, compared to the baseline ICIS, for several visual classifiers.

| Model | Method | Generalized Zero-Shot Setting | | | | Zero-Shot Setting | |
|---|---|---|---|---|---|---|---|
| | | Train@1 | Train@5 | Test@1 | Test@5 | Test@1 | Test@5 |
| ResNet50 | ICIS | **74.18** | **91.76** | 13.04 | 51.68 | 49.80 | 77.04 |
| | Ours | 66.77 | 89.34 | **33.28** | **66.20** | **50.88** | **77.62** |
| ResNet101 | ICIS | **76.05** | **92.86** | 11.18 | 43.26 | **46.06** | **73.48** |
| | Ours | 69.42 | 90.19 | **25.92** | **57.80** | 44.36 | 72.96 |
| WideResNet50 | ICIS | **78.29** | **93.57** | 11.82 | 49.08 | **51.00** | **79.26** |
| | Ours | 74.27 | 92.05 | **30.20** | **64.92** | 49.98 | 78.46 |
| WideResNet101 | ICIS | **78.89** | **93.67** | 9.26 | 46.02 | 46.38 | 75.22 |
| | Ours | 75.65 | 92.59 | **26.66** | **60.60** | **48.58** | **75.82** |
| ResNeXt50$_{32x4d}$ | ICIS | **77.54** | **93.32** | 8.78 | 44.00 | 44.24 | 74.52 |
| | Ours | 72.44 | 90.97 | **29.96** | **63.00** | **46.90** | **74.96** |
| MobileNetv3-L | ICIS | **74.22** | **91.53** | 3.30 | 21.44 | **39.56** | 65.56 |
| | Ours | 68.26 | 89.84 | **18.90** | **43.42** | 39.32 | **66.00** |
| ViT-B/16$_{swag}$ | ICIS | 85.43 | 97.11 | 4.98 | 50.62 | 57.72 | **84.18** |
| | Ours | **85.47** | **97.35** | **9.18** | **53.84** | **58.20** | 83.72 |
| BEiT-L/16 | ICIS | **87.79** | **98.32** | 5.72 | 49.62 | **56.96** | 78.84 |
| | Ours | 87.57 | 98.26 | **10.84** | **51.52** | 54.10 | **80.22** |
| DINOv2-B | ICIS | **81.30** | **96.08** | 13.96 | 44.70 | **46.60** | 72.14 |
| | Ours | 79.26 | 95.95 | **21.52** | **49.36** | 43.50 | **73.72** |
| ConvNeXtV2-B$_{pt-384}$ | ICIS | 87.89 | 98.20 | 6.06 | 48.64 | 56.36 | 76.74 |
| | Ours | **87.90** | **98.31** | **9.36** | **50.60** | **56.72** | **77.58** |

cases, in the zero-shot setting as well. As this is an ill-posed problem, the testing accuracies may not be particularly high. This phenomenon is standard in zero-shot learning (Xian et al., 2017a), especially on the challenging ImageNet dataset. Evaluation on other models are presented in Section F of the appendix. We also present an analysis on test prompt sensitivity in Section E of the appendix. In Figure 5, we present a t-sne visualization of the generated embeddings for new unseen classes taken from ImageNet-21K. As seen, semantically similar classes are clustered together (*e.g.,* water sports cluster, people cluster). After tuning the hyperparameters of the **MLP** mapper, we train it on the full set of classes (1, 000 for ImageNet) for further applications. In Section D of the appendix, we present ablation studies on Input Dropout and Soft Dropout. Our findings show that Input Dropout boosts top-1 zero-shot accuracy by 11.38% points, while Soft Dropout further boosts it by an additional 16.04% points. Therefore, these augmentations together provide a significant increase of 27.42% points in top-1 zero-shot accuracy. We also present ablation studies on different text encoders.

## 4.2 ZERO-SHOT NLES

**Baselines and Dataset:** We use the ImageNet-X (Sammani & Deligiannis, 2023) dataset which provides explanations for ImageNet categories, describing them with distinctive and physical features. It consists of 141K training samples, 2K for validation and 1K for testing. As our method is zero-shot, we only use the testing split for evaluation purposes. We compare against 4 supervised baseline NLE models which are trained explicitly to generate those explanations: NLX-GPT (Sammani et al., 2022) and Uni-NLX (Sammani & Deligiannis, 2023). The subscript *ft* means that the NLE model is preceded by pretraining on 1M image- caption pairs before being finetuned on ImageNet-X. All baselines are trained on the ImageNet-X training set.

**Metrics:** Supervised NLE models use natural language generation (NLG) metrics such as BLEU (Papineni et al., 2002) to measure the n-gram overlap between the generated NLE and the ground-truth one. We avoid evaluation using NLG metrics because it is challenging to expect an n-gram overlap between the ground-truth annotation and a zero-shot explanation. As a result, we resort to using

semantic-based evaluation metrics that, to the best of our ability, reflect faithfulness. Given a NLE for an image $x$, we use the following metrics: **CLIP-S:** We use the CLIP-Score (Hessel et al., 2021), a metric that leverages the powerful CLIP model as an external "judge" to assess the matching score between an image-text pair. Here, we use the CLIP-S to judge how associated the image $x$ is to its NLE. That is, `CLIP-S = CLIP(x, NLE)`. An NLE which truly reflects the image and its content, is given a high matching score. **LPIPS:** Perceptual Similarity (Zhang et al., 2018) is a learned metric trained with human input that has shown to correlate very well with human judgment. It measures the similarity between two images semantically using deep network features rather than focusing on pixel-level differences. To utilize this metric, we need two images. Taking advantage of the remarkable ability of current text-to-image models in generating realistic images, we first generate an image by re-formulating the NLE as a text prompt to the Stable Diffusion Text-to-Image model (Rombach et al., 2021) to obtain $x^g$ (see Section I of the appendix for more details on this). We then compare $x^g$ against the image $x$ using the LPIPS metric: `LPIPS(x, x^g)`. The LPIPS score is low when the two images share similar semantic features. The lower the LPIPS, the more the NLE truly reflects the visual features. We use the trained LPIPS models[1] based on AlexNet (A) and SqueezeNet (S). Finally, **Cosine** is the cosine similarity between the visual features of the model we interpret for the image $x$ and that for $x^g$: $\cos(M_V(x), M_V(x^g))$. When the NLE is correct and faithful, the synthesized image $x^g$ will activate the same visual features in the classifier as the original image $x$ that generated the NLE, resulting in a high cosine similarity score. Our method, including the 4 baselines follow exactly the same evaluation protocol, using the ImageNet-X test set.

We follow NLX-GPT and Uni-NLX where we generate both the prediction and explanation. As highlighted in NLX-GPT, this is important to ensure that the prediction and explanation come from the same reasoning process. Results are presented in Table 2. On all CNN models, our zero-shot NLEs outperform supervised NLEs on all metrics while remaining comparable on the cosine metric. On Transformer models, our zero-shot NLEs surpass the supervised NLEs on CLIP-S and perform comparably (and sometimes better) across other metrics. Note that the cosine metric only captures the geometric alignment (or direction) of these vectors in the Euclidean space. As a simple example, the cosine similarity between: $v_1 = [1, 1, 0]$ and $v_2 = [2, 2, 0]$, where $v_2 = 2v_1$ is, 1, indicating perfect alignment. Therefore, the cosine metric is suboptimal for capturing *semantically* relevant alignment. Evaluation on other models are presented in Section H of the appendix.

In Figure 6, we provide qualitative examples of images from different ImageNet categories. The first row presents NLEs that are expressive and free from contextual errors. For example, we can see that the rings presented on the snake's skin are distinctive of the category *king snake*. In the second row, we present examples where the NLEs are expressive but contain contextual errors. Despite these mistakes, they still help users understand the reasoning process. For example, the model identifies a *vine snake* based on its green, arrow-shaped head, and identifies a *tree frog* by its green, lizard-like head. We can also reveal that a ResNet-50 has associated *jellyfish* with "glowing". Upon inspecting the ImageNet training images for the class *jellyfish*, we observe that most of them feature glowing jellyfish. This suggests that the ResNet-50 model has taken a shortcut bias in learning this class. Additional qualitative examples are provided in Section J of the appendix.

### 4.3 ZERO-SHOT FINE-GRAINED CONCEPT DISCOVERY

We present one more human-friendly interpretable approach to deep learning models, which is fine-grained concept discovery where we aim to dissect the visual features into textual concepts in a zero-shot manner. In this context, we refer to concepts as textual descriptors; short descriptions in natural language that describe physical features of many objects in the world. This approach stems from how humans reason. For example, to justify an image as an *american robin*, we would describe the bird's beak, orange belly, and black back. In this approach, we are given a set of predefined concepts. We then query a visual classifier to determine whether these concepts are present in the classifier's visual features. Previously, this method was restricted to CLIP models (Menon & Vondrick, 2023). Now, with our approach, it can be applied to any visual classifier. Manually writing these textual concepts can be costly and does not scale to large class sets like ImageNet. To address this, we leverage large language models (LLMs), which demonstrate remarkable world knowledge across various domains, to generate these concepts. See Section K of the appendix for more details.

---

[1]https://github.com/richzhang/PerceptualSimilarity

Table 2: Quantitative Results on our zero-shot NLEs compared to supervised baselines, on several visual classifiers. Results are reported on the test set of ImageNet-X.

| Model | Metric | NLX-GPT | NLX-GPT$_{ft}$ | Uni-NLX | Uni-NLX$_{ft}$ | Ours |
|---|---|---|---|---|---|---|
| ResNet50 | CLIP-S ↑ | 28.53 | 28.52 | 28.38 | 28.52 | **30.56** |
|  | LPIPS(A) ↓ | 0.721 | 0.719 | 0.720 | 0.716 | **0.706** |
|  | LPIPS(S) ↓ | 0.624 | 0.623 | 0.620 | 0.620 | **0.614** |
|  | Cosine ↑ | 0.703 | 0.704 | 0.703 | **0.710** | 0.708 |
| ResNeXt50$_{32x4d}$ | CLIP-S ↑ | 28.53 | 28.52 | 28.38 | 28.52 | **30.73** |
|  | LPIPS(A) ↓ | 0.717 | 0.717 | 0.720 | 0.719 | **0.707** |
|  | LPIPS(S) ↓ | 0.621 | 0.620 | 0.622 | 0.623 | **0.616** |
|  | Cosine ↑ | **0.680** | 0.677 | 0.677 | 0.677 | 0.676 |
| DenseNet161 | CLIP-S ↑ | 28.53 | 28.52 | 28.38 | 28.52 | **30.56** |
|  | LPIPS(A) ↓ | 0.717 | 0.718 | 0.718 | 0.723 | **0.706** |
|  | LPIPS(S) ↓ | 0.620 | 0.620 | 0.620 | 0.624 | **0.616** |
|  | Cosine ↑ | 0.589 | **0.591** | 0.589 | 0.588 | 0.588 |
| DINOv2-B | CLIP-S ↑ | 28.53 | 28.52 | 28.38 | 28.52 | **30.15** |
|  | LPIPS(A) ↓ | 0.719 | 0.716 | 0.717 | 0.716 | **0.712** |
|  | LPIPS(S) ↓ | 0.622 | 0.621 | **0.619** | **0.619** | **0.619** |
|  | Cosine ↑ | **0.362** | 0.360 | 0.354 | 0.356 | 0.314 |
| ViT-B/16$_{pt}$ | CLIP-S ↑ | 28.53 | 28.52 | 28.38 | 28.52 | **30.92** |
|  | LPIPS(A) ↓ | 0.723 | 0.723 | 0.717 | 0.722 | **0.710** |
|  | LPIPS(S) ↓ | 0.624 | 0.626 | **0.619** | 0.624 | 0.621 |
|  | Cosine ↑ | **0.450** | 0.440 | 0.447 | 0.445 | 0.425 |

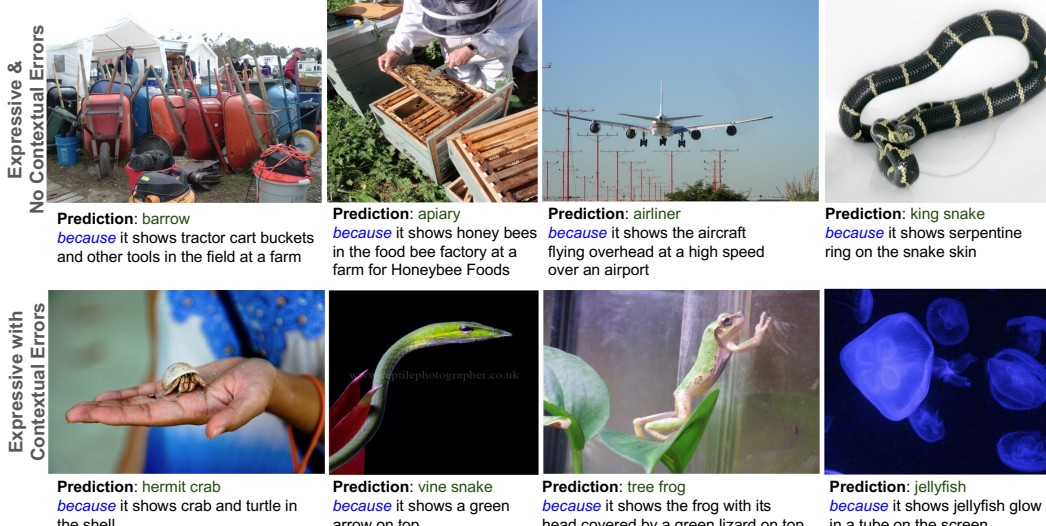

Figure 6: Qualitative examples of our zero-shot NLEs on ImageNet for a ResNet-50 classifier.

Performing this application with our method is simple, and is illustrated in Figure 3d. Given a large set of concepts $U$ with $|U| = Z$, every concept $u_j$ is encoded using the text encoder $M_T$ followed by the **MLP** to yield a concept embedding $\hat{\mathbf{w}}_{u_j}$. That is, $\hat{\mathbf{w}}_{u_j} = \mathbf{MLP}(\mathbf{M_T}(u_j))$. We then compute the cosine similarity between the visual features $f$ and $\hat{\mathbf{w}}_{u_j}, \forall j = 1, \ldots, Z$. We then take the top-$B$ scoring concepts. Qualitative results are shown in Figure 7 for different models. This allows us to reveal what concepts different model features encode. Interestingly, we see that the BeiT-L models assigns the top-concept for the "tench" prediction as "10 legs". This classification arises from the patchification process of the transformer, where multiple patches comprise of the fins of the fish, which are identified as "10 legs" due to their physical appearance.

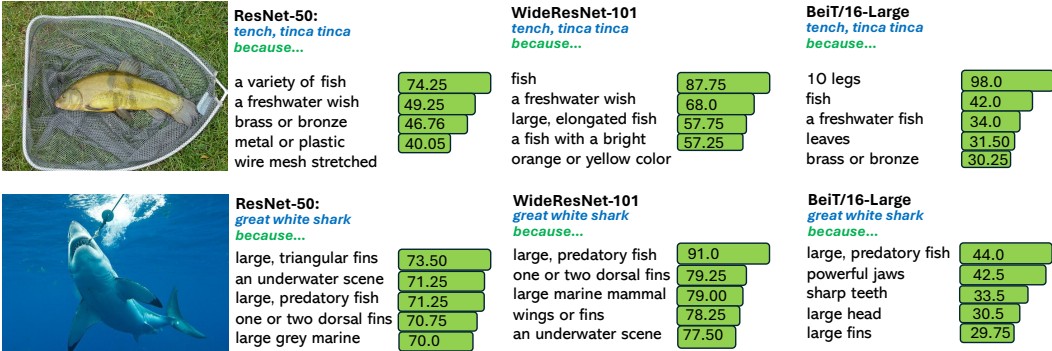

Figure 7: Qualitative examples of concept discovery on ImageNet for different classifiers. The cosine similarity is scaled by 2.5 for all models, in order to stretch the range of the score distribution to [0, 1]

Finally, we conduct a quantitative evaluation of the detected concepts to assess their effectiveness and faithfulness to the visual features. We use the top-detected concepts as supplementary information to regress the class embeddings. If the detected concepts are faithful, they should lead to a better regression of the class embeddings, and as a result a higher accuracy. Our baseline uses only the class name to regress the class embeddings. That is, $\hat{\mathbf{w}}_{c_i} = \mathbf{MLP}(\mathbf{M_T}(t_i))$ where $t_i$ is: *an image of a {class name}*. For the evaluation, we change $t_i$ to $t_{ic}$ which now include both the class name *and* the detected concepts: $t_{ic}$ = *an image of a {class name, $u_b$}* where $u_b$ is a concept from the top-$B$ detected concepts. Results on ImageNet are presented in Table 3 on several models. As shown, the detected concepts lead to significant improvements in accuracy, which shows the effectiveness and faithfulness of the concepts to the visual features. In this experiment, we use $B = 15$ and average the text encoder outputs to a single feature vector, before we feed it as input to the **MLP**. We also found that setting $B = 10$ yields to approximately the same results (*e.g.,* 73.61 Top-1 accuracy for a ResNet-50). Evaluation on other models are provided in Section G of the appendix.

## 4.4 ADDITIONAL APPLICATIONS

In Section B of the appendix, we also present zero-shot image captioning experiments on the COCO dataset (Lin et al., 2014) as an additional application of our method, achieving new state-of-the-art results while also showing the generalizability of our method to datasets beyond ImageNet. Furthermore, we present experiments on zero-shot transfer to other datasets in Section C of the appendix.

## 5 CONCLUSION

We proposed a zero-shot faithful NLE method for visual classifiers. We also presented other applications on zero-shot image classification and fine-grained concept discovery. Note that our work offers a solution to a wide range of zero-shot applications that were previously restricted to CLIP models. Our work removes this limitation, and can now be applied to any visual classifier. Additionally, our work offers a method for annotating sparse autoencoder latents trained on any visual classifier. Finally, as with any research work, our method is accompanied by its own set of limitations, which we discuss in Section A of the appendix.

Table 3: Quantitative results of concept discovery

| Model | Baseline | | + Concepts | |
|---|---|---|---|---|
| | Top-1 | Top-5 | Top-1 | Top-5 |
| ResNet50 | 63.42 | 87.02 | **73.66** | **92.18** |
| ResNet101 | 65.07 | 86.95 | **75.32** | **92.91** |
| WideResNet50$_{v2}$ | 73.51 | 90.10 | **81.06** | **95.64** |
| WideResNet101$_{v2}$ | 75.25 | 90.60 | **82.05** | **95.95** |
| ResNeXt101$_{64x4d}$ | 75.52 | 90.79 | **82.85** | **96.41** |
| DenseNet161 | 67.84 | 88.58 | **75.37** | **93.11** |
| EfficientNetv2-S | 76.64 | 91.99 | **84.00** | **96.79** |
| EfficientNetv2-M | 77.82 | 92.37 | **85.03** | **97.17** |
| ShuffleNetv2$_{x2.0}$ | 68.02 | 86.87 | **75.04** | **92.64** |
| MobileNetv3-L | 63.32 | 85.20 | **71.76** | **90.62** |
| ViT-B/32 | 68.66 | 85.76 | **75.31** | **92.34** |
| ViT-B/16$_{swag}$ | 77.84 | 93.00 | **84.42** | **97.48** |
| ViT-L/32 | 69.83 | 86.66 | **76.47** | **92.97** |
| ViT-L/16 | 72.35 | 88.58 | **79.44** | **94.51** |
| Swin-S | 75.40 | 90.66 | **82.45** | **96.34** |
| ViT-B/16$_{pt}$ | 76.90 | 92.77 | **82.40** | **96.79** |
| BeiT-B/16 | 77.27 | 92.47 | **84.56** | **97.37** |
| BEiT-L/16 | 79.90 | 93.58 | **87.06** | **98.22** |
| DINOv2-B | 73.49 | 91.29 | **81.99** | **96.37** |
| CvT-21 | 74.16 | 90.10 | **81.04** | **95.15** |
| ConvNeXt-B | 76.54 | 91.66 | **83.72** | **96.79** |
| ConvNeXt-B$_{pt}$ | 77.89 | 92.88 | **84.98** | **97.66** |
| ConvNexTv2-B | 77.31 | 91.83 | **84.58** | **97.11** |
| ConvNexTv2-B$_{pt}$ | 78.93 | 93.32 | **85.96** | **97.87** |
| ConvNeXtv2-B$_{pt-384}$ | 80.05 | 93.54 | **87.28** | **98.32** |

ACKNOWLEDGEMENT

Fawaz Sammani is fully and solely funded by the Fonds Wetenschappelijk Onderzoek (FWO) (PhD fellowship strategic basic research 1SH7W24N). N. Deligiannis acknowledges support from the Francqui Foundation (2024-2027 Francqui Research Professorship on Trustworthy AI) and the "Onderzoeksprogramma Artificiele Intelligentie (AI) Vlaanderen" programme.

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

## A    LIMITATIONS AND ETHICAL CONCERNS

All research has inherent limitations, and this study is no exception. It is important to acknowledge these constraints. We highlight two limitations of our method.

Our method may occasionally tend to focus on the general image features rather than specific ones. Figure 8 illustrates such cases. In the first example, the "jars" in the background influence the model to produce content related to "caffeine" and "garlic", likely due to their proximity in the embedding space to the word "jars." In another instance, the upper text above the puzzle in the image resembles a list of answers, leading the model to generate content associated with "tests." Additionally, the model hallucinates that the test is related to computers. Note that when the image is solely composed of the main object without other distractions such as the background, the generated NLE is usually meaningful (*e.g.,* an image from the same "crossword puzzle" class in Figure 9). Our method can also potentially generate inappropriate language, as seen in the final example. This occurs due to the proximity of certain inappropriate terms to the word "cock" in the embedding space, which refers to the bird resembling the crane in the image, but can also refer to the male's genitalia.

Another limitation of our approach is its slower performance compared to supervised NLE models. Our method relies on an inference-time approach, which results in the generation of a single NLE taking approximately 35 seconds on a single RTX 3090 GPU. In contrast, supervised NLE models are trained once, allowing for inference to occur in a single forward pass, making our approach more time-consuming. Although this is the case, we emphasize that our approach offers greater flexibility, operates effectively in an open-set environment, and can be adapted to any example or classifier on-the-fly. We prioritize these key factors as they are more important in NLEs, thereby outbalancing the increased explanation generation time.

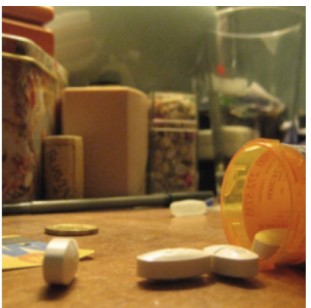 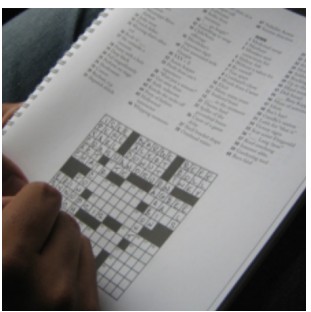 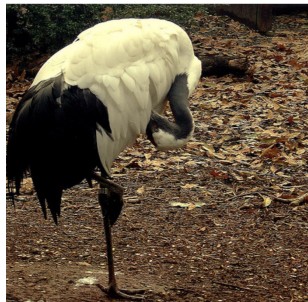

*prediction*: pill_bottle
*because...*
it shows caffeine glass jars and a
teaspoon of garlic powder in the spice jar

*prediction*: crossword puzzle
*because...*
it shows the answers to test questions
on the basics of a computer

*prediction*: crane
*because...*
it shows the cock p***s d**k of a
crane

Figure 8: Qualitative examples of failure cases (first two examples) and inappropriate language (last example) for NLEs produced.

## B    ZERO-SHOT IMAGE CAPTIONING

We present zero-shot image captioning as an additional application. We use the MLP trained to synthesize the full 1,000 ImageNet class weights, and apply the method presented in Section 3.2 on images from the COCO dataset (Lin et al., 2014). We report results on the common "Karpathy test split" benchmark using various vision classifiers. As baselines, we compare against the current state-of-the-art systems on zero-shot image captioning, namely ZeroCap (Tewel et al., 2021) and ConZIC(Zeng et al., 2023), both which use CLIP. We use the common NLG metrics for evaluation: BLEU-4 (B@4) (Papineni et al., 2002), METEOR (M) (Banerjee & Lavie, 2005), CiDEr (C) (Vedantam et al., 2014) and SPICE (S) (Anderson et al., 2016). Results are presented in Table 4. Decoding features from BEiT-L achieves state-of-the-art results on CiDEr and SPICE, which are the two most important metrics for evaluating image captioning systems. ViT-B/16$_{pt}$ performs best on METEOR, while the baseline ZeroCap leads in BLEU-4. Notably, even with a basic ResNet-50 vision encoder trained on ImageNet-1K of 1.2 million images, our approach surpasses baseline methods on CiDEr and SPICE and achieves comparable results on other metrics, despite the fact that the baseline methods use the far more powerful CLIP vision encoder, trained on 400 million image-text pairs.

We also find that Transformer-based models offer an advantage over CNNs on this task. Note that the COCO dataset differs in distribution from ImageNet in terms of context and object composition in images, demonstrating the robust transferability of our method to new datasets in a completely zero-shot manner.

Table 4: Zero-Shot Image Captioning Results on MSCOCO Karpathy Test Split

| Method | B@4 | M | C | S |
|---|---|---|---|---|
| ZeroCap | **2.6** | 11.5 | 14.6 | 5.5 |
| ConZIC | 1.3 | 11.5 | 12.8 | 5.2 |
| **Ours** | | | | |
| ResNet50 | 1.7 | 11.2 | 14.9 | 6.6 |
| ResNet101 | 1.8 | 11.3 | 15.5 | 6.6 |
| DINOv2-B | 1.8 | 11.6 | 16.2 | 7.1 |
| ConvNeXtV2-B$_{pt-384}$ | 2.0 | 11.5 | 16.9 | 7.2 |
| ViT-B/16$_{pt}$ | 1.7 | **11.9** | 17.4 | 7.4 |
| BEiT-L/16 | 1.8 | 11.7 | **17.6** | **7.6** |

## C  ZERO-SHOT TRANSFER

We investigate the performance of our method in zero-shot transfer, a scenario where a model trained on one dataset can generalize to other datasets without additional training. Specifically, we use the MLP trained to synthesize the 1,000 ImageNet class weights and test its ability to synthesize class weights for other datasets. For this evaluation, we select the Places365 dataset (Zhou et al., 2017) specializing in scene classification, and the DTD dataset (Cimpoi et al., 2014) specializing in texture type classification. These datasets are particularly challenging; for instance, the powerful CLIP model (Radford et al., 2021), trained on 400 million image-text pairs, achieves only 41.7% top-1 zero-shot accuracy on DTD and 37.37% on Places365. We report the Test@1 and Test@5 Generalized Zero-Shot Accuracy on the full validation sets of the selected datasets. Notably, in this context, the generalized zero-shot accuracy on the test data is equivalent to that in the zero-shot setting, as the evaluation encompasses all unseen classes in the Places365 and DTD datasets. It is important to note that the objective of this experiment is not to surpass CLIP's performance, as our method is fundamentally different and not directly comparable with CLIP. Unlike CLIP, our approach is trained on text data only, without any image data, and utilizes only 1,000 samples—400,000× fewer data than CLIP. Instead, the goal of this experiment is to demonstrate the zero-shot transfer capabilities of our method. Results are presented in Table 5. We observe that state-of-the-art models leveraging large-scale pretraining significantly outperform other models on this task, with BEiT-L achieving the best results. We also observe a trend consistent with the existing out-of-distribution (OOD) literature: models leveraging large-scale pretraining, particularly Transformers, demonstrate superior OOD performance. This pattern holds true across both datasets.

Table 5: Zero-Shot Transfer Performance of different models to Places365 and DTD datasets

| Model | Places365 | | DTD | |
|---|---|---|---|---|
| | Test@1 | Test@5 | Test@1 | Test@5 |
| ResNet50 | 10.76 | 30.92 | 14.36 | 32.29 |
| WideResNet101 | 13.02 | 33.89 | 12.77 | 31.97 |
| ViT-B/32 | 13.89 | 34.41 | 15.74 | 33.24 |
| ViT-L/16 | 14.17 | 33.91 | 15.53 | 32.39 |
| ConvNeXt-Base | 15.03 | 35.92 | 18.46 | 34.79 |
| DINOv2-Base | 16.66 | 40.88 | 17.66 | 40.05 |
| ViT-B/16 (pt) | 18.46 | 44.53 | 15.85 | 33.30 |
| ConvNeXtV2-B (pt-384) | 18.48 | 42.44 | 19.20 | 35.96 |
| BEiT-L/16 | **19.18** | **43.59** | **20.27** | **37.13** |

## D  ABLATION STUDIES

We present ablation studies on learning class embeddings, demonstrating that each component contributes non-trivially. In Table 6, we start by showing results on simple baselines to ensure that our method is not purely statistical and demonstrates effective learning. For this experiment, we use the ResNet-50 model and report the top-1 and top-5 accuracy for the test split used in Section 4.1. As shown, when the class embedding of the test set is consistently set to the mean of all class embeddings from the training set, the **MLP** fails. We also include a baseline where random values are sampled from the minimum and maximum range of the training class embeddings, again showing that the **MLP** fails. The same outcome occurs when using randomly initialized parameters for the **MLP**. Our main baseline uses the MSE distance loss to learn class embeddings. Replacing the MSE with a cosine loss which incorporates angular information improves top-1 accuracy by

Table 6: Ablation studies of the method in Section 4.1, using ResNet-50

| Ablation | Top-1 | Top-5 |
|---|---|---|
| *Learning Class Embeddings* | | |
| Mean of Train Set | 0.00 | 0.00 |
| Random Values | 0.00 | 0.20 |
| Random Parameters | 0.20 | 0.62 |
| Baseline (MSE Loss) | 3.12 | 35.56 |
| Cosine Loss | 5.86 | 40.64 |
| + Input Drop | 17.24 | 54.38 |
| + Soft Dropout | **33.28** | **66.20** |
| *Different text encoders* | | |
| MPNet-B-QA$_{v1}$ | 24.16 | 57.50 |
| DistilRoberta$_{v1}$ | 25.34 | 57.62 |
| MiniLM-L12$_{v1}$ | 26.94 | 54.74 |
| MPNet-Base$_{v2}$ | **33.28** | **66.20** |

approximately 3 points. Applying dropout to the input increases top-1 accuracy by 11 points, and the addition of learned soft dropout significantly enhances performance by 16 points. Next, we ablate different text encoders from Reimers & Gurevych (2019) using the Sentence Transformers library[2]. As shown, the MPNet-Base$_{v2}$ (Song et al., 2020) provides the best results.

## E  PROMPT ANALYSIS

In this section, we analyze the impact of prompt templates used at inference time. CLIP models are known to be sensitive to prompt variations (e.g., adding the letter "a" can significantly improve accuracy, as shown in Zhou et al. (2021)). To explore this, we evaluated our pretrained MLP for ResNet50—trained with the prompt "an image of a class"—on the Generalized Zero-Shot accuracy for unseen test classes using various prompts in the text encoder. Results are presented in Table 7 for several prompts, arranged in increasing order of sensitivity. Our findings reveal that, like CLIP models, the MLP is also sensitive to prompt variations. Adding or changing one word can result in degradation of zero-shot accuracy. However, this issue can be resolved by averaging the results of all prompts, similar to the approach used by CLIP.

Table 7: Performance with different prompts at inference

| Prompt | Test@1 | Test@5 |
|---|---|---|
| an image of a {class} | 33.28 | 66.20 |
| an image of the {class} | 32.92 | 65.02 |
| an image of one {class} | 31.98 | 64.06 |
| an image of a large {class} | 31.52 | 63.86 |
| an image of a nice {class} | 31.18 | 64.86 |
| an image of a weird {class} | 30.90 | 64.24 |
| a cropped image of a {class} | 30.22 | 63.42 |
| a black and white image of the {class} | 30.14 | 61.74 |

---

[2]https://sbert.net/

## F ADDITIONAL MODELS ON EVALUATING LEARNED CLASS EMBEDDINGS

In this section, we provide quantitative evaluation of the learned class embeddings for additional models not included in the main manuscript, and compare them with the baseline ICIS. Results are provided in Tables 8 and 9. Our method consistently outperforms ICIS by a large margin on the test set in the generalized zero-shot accuracy settings, and in most cases, in the generalized setting as well.

Table 8: Quantitative Results for additional models on evaluating class embeddings via zero-shot image classification on ImageNet-1K, compared to the baseline ICIS, for several visual classifiers

| Model | Method | Generalized Zero-Shot Setting | | | | Generalized Setting | |
|---|---|---|---|---|---|---|---|
| | | Train@1 | Train@5 | Test@1 | Test@5 | Test@1 | Test@5 |
| ResNet50$_{v2}$ | ICIS | **80.80** | **95.17** | 3.72 | 37.94 | **44.30** | **68.08** |
| | Ours | 79.65 | 94.76 | **11.66** | **44.94** | 44.24 | 67.84 |
| ResNet101$_{v2}$ | ICIS | **82.09** | **95.70** | 1.30 | 28.60 | 39.24 | 63.52 |
| | Ours | 81.41 | 95.50 | **6.10** | **37.80** | **40.28** | **64.76** |
| WideResNet50$_{v2}$ | ICIS | **81.56** | **95.51** | 1.32 | 30.34 | 41.32 | 65.58 |
| | Ours | 80.49 | 95.33 | **10.66** | **43.04** | **42.82** | **65.88** |
| WideResNet101$_{v2}$ | ICIS | **82.99** | 95.78 | 0.84 | 32.86 | 43.04 | 66.60 |
| | Ours | 82.62 | **95.80** | **8.98** | **43.78** | **44.96** | **68.88** |
| ResNeXt50$_{32x4d, v2}$ | ICIS | **81.72** | **95.31** | 1.78 | 31.98 | **40.30** | 66.00 |
| | Ours | 81.07 | 95.06 | **6.16** | **40.66** | 39.42 | **67.92** |
| ResNeXt101$_{64x4d}$ | ICIS | **83.13** | **96.11** | 0.48 | 23.18 | 38.86 | 68.82 |
| | Ours | 83.02 | 96.00 | **8.00** | **43.86** | **40.32** | **72.34** |
| ResNeXt101$_{32x8d}$ | ICIS | **79.18** | **94.16** | 11.36 | 45.64 | **49.06** | 73.82 |
| | Ours | 75.88 | 92.96 | **27.44** | **60.78** | 48.84 | **73.88** |
| ResNeXt101$_{32x8d, v2}$ | ICIS | **82.95** | **95.98** | 2.56 | 30.32 | 41.88 | 66.96 |
| | Ours | 82.32 | 95.91 | **11.46** | **44.12** | **44.12** | **70.04** |
| DenseNet121 | ICIS | **72.24** | **90.84** | 9.40 | 41.84 | 43.30 | **69.90** |
| | Ours | 69.08 | 89.99 | **19.24** | **49.24** | **44.38** | 68.04 |
| DenseNet161 | ICIS | **76.07** | **93.28** | 9.16 | 42.08 | 45.42 | 73.46 |
| | Ours | 73.00 | 92.54 | **21.44** | **52.88** | **45.88** | **74.26** |
| EfficientNetv2-S | ICIS | **84.53** | **96.72** | 5.78 | 48.04 | 52.78 | 76.50 |
| | Ours | 84.35 | 96.62 | **7.24** | **50.40** | **56.90** | **80.18** |
| EfficientNetv2-M | ICIS | **85.38** | 96.94 | 7.10 | 50.36 | **56.18** | 77.16 |
| | Ours | 85.31 | **96.99** | **10.40** | **50.82** | 53.00 | **77.70** |
| ShuffleNetv2$_{x2.0}$ | ICIS | **76.27** | **92.92** | 1.44 | 26.28 | 39.40 | 70.42 |
| | Ours | 74.74 | 92.29 | **7.56** | **38.12** | **40.12** | **70.98** |
| ConvNext-Tiny | ICIS | **82.78** | **96.16** | 2.34 | 25.94 | 33.12 | **57.14** |
| | Ours | 82.33 | 96.11 | **2.78** | **26.48** | **33.48** | 56.20 |
| ConvNext-Small | ICIS | **84.26** | 96.59 | 3.56 | **38.10** | **44.90** | 68.96 |
| | Ours | 84.13 | **96.64** | **4.76** | 37.70 | 43.54 | **69.08** |
| ConvNext-Base | ICIS | **84.58** | **96.82** | 4.72 | 43.22 | 49.26 | 71.98 |
| | Ours | 84.44 | 96.80 | **5.50** | **45.40** | **51.50** | **74.20** |

## G ADDITIONAL MODELS FOR EVALUATING CONCEPT DISCOVERY

In this section, we provide quantitative evaluation of concept discovery for additional models not included in the main manuscript. Results are provided in Table 10. As shown, the detected concepts

Table 9: Quantitative Results for additional models on evaluating class embeddings via zero-shot image classification on ImageNet-1K, compared to the baseline ICIS, for several visual classifiers

| Model | Method | Generalized Zero-Shot Setting | | | | Generalized Setting | |
|---|---|---|---|---|---|---|---|
| | | Train@1 | Train@5 | Test@1 | Test@5 | Test@1 | Test@5 |
| ViT-B/32 | ICIS | **76.12** | 92.16 | 2.22 | 26.98 | **35.60** | **55.10** |
| | Ours | 75.99 | **92.25** | **2.66** | **27.42** | 33.46 | 53.52 |
| ViT-L/32 | ICIS | **77.35** | 92.89 | 1.80 | 28.20 | 36.70 | 58.80 |
| | Ours | 77.31 | **93.01** | **2.52** | **29.50** | **36.98** | **58.82** |
| ViT-L/16 | ICIS | 79.98 | **94.54** | **3.76** | 34.10 | 44.98 | 66.82 |
| | Ours | **80.14** | 94.45 | 2.22 | **35.80** | **46.62** | **67.92** |
| Swin-T | ICIS | **81.29** | 95.58 | 1.74 | 27.78 | 38.76 | **65.06** |
| | Ours | 81.20 | **95.63** | **2.40** | **30.26** | **38.86** | 64.98 |
| Swin-S | ICIS | **83.30** | **96.38** | 4.30 | 38.12 | **42.56** | **66.22** |
| | Ours | 83.17 | 96.35 | **5.46** | **39.48** | 41.64 | 64.32 |
| ViT-B/16$_{pt}$ | ICIS | **83.82** | **96.71** | 16.80 | 53.70 | 51.46 | 80.48 |
| | Ours | 83.21 | 96.67 | **20.08** | **57.66** | **52.22** | **81.62** |
| BeiT-B/16 | ICIS | 85.18 | 97.34 | 3.70 | 42.72 | 49.72 | 78.32 |
| | Ours | **85.20** | **97.50** | **5.94** | **47.14** | **53.02** | **79.20** |
| DINOv2-S | ICIS | **77.17** | **94.24** | 17.22 | 44.54 | 43.58 | 69.82 |
| | Ours | 76.08 | 94.14 | **21.42** | **49.02** | **46.56** | **71.22** |
| CvT-21 | ICIS | **81.99** | **95.29** | 2.50 | **45.66** | **47.80** | 75.58 |
| | Ours | 81.98 | 95.14 | **3.78** | 44.78 | 47.68 | **75.72** |
| ConvNexT-B$_{pt}$ | ICIS | **85.77** | 97.55 | 6.72 | 42.46 | 50.08 | 72.72 |
| | Ours | 85.59 | **97.64** | **8.56** | **49.98** | **53.48** | **76.12** |
| ConvNexTv2-B | ICIS | **85.38** | 96.99 | 4.28 | 41.16 | 44.98 | **67.62** |
| | Ours | 85.30 | **97.13** | **5.42** | **44.08** | **46.66** | 66.92 |
| ConvNexTv2-B$_{pt}$ | ICIS | **86.77** | 97.77 | 6.08 | 49.16 | 56.34 | **78.46** |
| | Ours | 86.61 | **97.88** | **9.78** | **52.30** | **56.38** | 77.98 |

lead to significant improvements in accuracy over the baseline, which shows the effectiveness of those concepts, and that they are faithful to the visual features.

Table 10: Quantitative Results for additional models on concept discovery

| Model | Baseline | | +Concepts | |
|---|---|---|---|---|
| | Top-1 | Top-5 | Top-1 | Top-5 |
| ResNet50$_{v2}$ | 72.85 | 89.78 | **80.25** | **95.31** |
| ResNet101$_{v2}$ | 73.88 | 89.73 | **81.40** | **95.76** |
| WideResNet50 | 69.86 | 89.34 | **77.60** | **93.70** |
| WideResNet101 | 70.75 | 89.39 | **77.97** | **93.95** |
| ResNeXt50$_{32x4d}$ | 68.19 | 88.17 | **76.22** | **93.27** |
| ResNeXt50$_{32x4d, v2}$ | 73.58 | 89.62 | **80.70** | **95.26** |
| ResNeXt101$_{32x8d}$ | 71.04 | 89.74 | **78.44** | **94.21** |
| ResNeXt101$_{32x8d, v2}$ | 75.24 | 90.73 | **82.39** | **96.22** |
| DenseNet121 | 64.09 | 85.91 | **71.10** | **90.91** |
| ConvNext-T | 74.38 | 89.15 | **81.87** | **96.07** |
| ConvNext-S | 76.20 | 90.75 | **83.23** | **96.65** |
| Swin-T | 73.32 | 89.09 | **80.66** | **95.58** |
| DINOv2-S | 70.61 | 89.62 | **76.94** | **94.45** |

## H  ADDITIONAL MODELS FOR EVALUATING NLEs

In this section, we provide quantitative evaluation of NLEs for additional models not included in the main manuscript. Results are provided in Table 11. Our zero-shot NLEs outperform the supervised baselines on the CLIP-S metric, while achieving comparable or, in many cases, surpassing scores on other metrics.

Table 11: Quantitative Results for additional models on our zero-shot NLEs compared to supervised baselines. Results are reported on the test set of ImageNet-X.

| Model | Metric | NLX-GPT | NLX-GPT$_{ft}$ | Uni-NLX | Uni-NLX$_{ft}$ | Ours |
|---|---|---|---|---|---|---|
| WideResNet50 | CLIP-S ↑ | 28.53 | 28.52 | 28.38 | 28.52 | **30.70** |
| | LPIPS(A) ↓ | 0.719 | 0.718 | 0.717 | 0.715 | **0.710** |
| | LPIPS(S) ↓ | 0.622 | 0.623 | **0.618** | 0.620 | 0.620 |
| | Cosine ↑ | **0.649** | 0.641 | 0.642 | 0.647 | 0.635 |
| WideResNet101 | CLIP-S ↑ | 28.53 | 28.52 | 28.38 | 28.52 | **30.66** |
| | LPIPS(A) ↓ | 0.719 | 0.720 | 0.718 | 0.718 | **0.710** |
| | LPIPS(S) ↓ | 0.622 | 0.624 | 0.620 | 0.623 | **0.618** |
| | Cosine ↑ | **0.642** | 0.640 | 0.641 | 0.640 | 0.633 |
| ConvNeXtV2-B$_{pt}$ | CLIP-S ↑ | 28.53 | 28.52 | 28.38 | 28.52 | **29.83** |
| | LPIPS(A) ↓ | 0.720 | 0.718 | 0.719 | 0.717 | **0.716** |
| | LPIPS(S) ↓ | 0.623 | 0.621 | 0.623 | **0.620** | 0.626 |
| | Cosine ↑ | **0.458** | 0.443 | 0.444 | 0.445 | 0.364 |
| BeiT-L/16 | CLIP-S ↑ | 28.53 | 28.52 | 28.38 | 28.52 | **30.04** |
| | LPIPS(A) ↓ | 0.723 | 0.718 | **0.715** | 0.716 | 0.716 |
| | LPIPS(S) ↓ | 0.623 | 0.619 | 0.620 | **0.618** | 0.622 |
| | Cosine ↑ | **0.440** | 0.429 | 0.427 | 0.432 | 0.372 |

## I  IMPLEMENTATION DETAILS

**Text-to-Image Model:** For the semantic-based evaluation, we use the Stable Diffusion v1.5 model (Rombach et al., 2021) from Hugging Face Diffusers library (von Platen et al., 2022), and precede the text prompt by the word `realistic`, following best practices from the text-to-image community. We perform this for the evaluation of both our method and the baselines. The NLE consists of concepts or features in text format that describe the visual features (e.g., "serpentine ring on snake skin") and formed as a natural language expression. Therefore, they can be formulated as a prompt that the text-to-image model can understand (e.g., "a realistic image showing serpentine rings on the snake skin."). Therefore, the synthesized image will depict an image incorporating these features.

**Learning Class Embeddings:** We begin by reminding readers that $k$ represents the dimension of the text features $s_i$ produced by the text encoder $M_T$, and $d$ is the dimensions of the class embeddings (and the visual features $f$). Our **MLP** is composed of two layers: $W_1$ and $W_2$. Each layer is followed by a layer normalization (Ba et al., 2016). At the first layer, we use a GELU activation function (Hendrycks & Gimpel, 2016) followed by a dropout layer with a probability of $0.5$. When using the learned soft dropout (see Section N), $W_1 \in \mathbb{R}^{k \times 2k}$ and $W_2 \in \mathbb{R}^{2k \times d}$. Otherwise, $W_1 \in \mathbb{R}^{k \times 4d}$ and $W_2 \in \mathbb{R}^{4d \times d}$. Note that the GLU activation function implementing the learned soft dropout also uses layer normalization, which we found to be important.

The MLP is trained with full batch gradient descent for 2500 epochs using the Adam Optimizer (Kingma & Ba, 2015) with a learning rate of 5e-3 and a cosine annealing schedule (Loshchilov & Hutter, 2017). On a single RTX3090 GPU, it takes roughly 10 seconds to train. All pretrained classifiers are obtained from the torchvision library. The ImageNet-21K pretrained ViT and ConvNeXt models, as well as all ConvNeXtv2 models are obtained from the timm library. The BeiT, DiNOv2 and CvT models are obtained from the huggingface library. We take the ImageNet class names provided by caffe-tensorflow.

**Generating NLEs:** We use the smallest GPT-2 (Radford et al., 2019) model of 124M parameters as our pretrained language model. The number of learnable prefixes is set to $5$ in each attention block of the GPT-2 model. As mentioned previously, the prefixes are set as key-value pairs such that each generated word in the explanation can attend to these prefixes. For example, if we have $Y$ currently generated words, then at each timestep, the number of keys and value tokens in the attention mechanism will always be $(5 + Y)$, while the number of query tokens will be $Y$. Consequently, the $Y$ query tokens will always attend to both the $Y$ key-value tokens *and* the 5 prefix tokens, resulting in a total of $(5 + Y)$ tokens to attend to. We train the prefixes with the Adam optimizer (Kingma & Ba, 2015) with a learning rate of $0.01$ and a weight decay of $0.3$ with a cosine annealing schedule for $I = 20$ iterations. Each iteration yields a generated sentence, and we select the sentence with the best CLIP-Score as the final NLE. The number of $K$ tokens sampled at each timestep is set to $512$. We use a maximum NLE length of $20$. We follow Tewel et al. (2023) and sample one hard prompt at each iteration, as contextual information for the language model, from the following list: ['Image of', 'Picture of', 'Photo of', 'Image shows', 'Picture shows', 'Photo shows', 'Image showing', 'Picture showing', 'Photo showing']. It is important to clarify that our task is not traditional image captioning, as we do not train the model to explicitly map visual features to annotated captions from a dataset. Instead, it can be regarded as *faithful* image captioning, which is also faithful NLEs, since we decode the visual features sourced from the model's internal representation, into language. To maintain the language model's coherent and fluent generation, we add the fluency loss from Tewel et al. (2021) with a weight of $0.8$. This is the cross-entropy loss between the language model distribution $o_l$ at timestep $l$ with the prefix and without the prefix. At each decoding step, we apply several modifications to the model's output probabilities: we discourage token repetition (Keskar et al., 2019) by reducing the scores of repetitive words by a factor of $2.0$. We also prevent the generation of repeated n-grams of order 3 to avoid repetitive phrases, by setting their score to negative infinity which eliminates those tokens from further sampling. We also enforce a minimum sequence length of 10 tokens by setting the `<.>` token score to 0, in order to prevent premature termination. We also exclude specific tokens such as unneeded character symbols, from being generated. Finally, as the sequence length approaches the target length, we promote ending it by increasing the score of the `<.>` token.

## J  ADDITIONAL QUALITATIVE EXAMPLES

In Figure 9, we provide additional qualitative examples of the NLEs generated for a ResNet-50 classifier. It is interesting to see that in the first example from the first row, we can infer that the model has encoded in its features the association of the"harvestman" spider with leaves and grassy areas—an insight that could not be uncovered using any other attribution technique. We can also see from the first example in the second row that the model associated the prediction of a "toucan" with the bird's colorful parts.

## K  GENERATING DESCRIPTORS FOR CONCEPT DISCOVERY WITH LLMS

We directly use the textual concepts provided by Menon & Vondrick (2023). This work uses GPT-3.5 for generating the concepts, using the following prompt: "What are useful visual features for distinguishing a {category name} in a photo?", where {category name} is replaced by the class name in the dataset. Additionally, the work utilizes in-context examples to guide the LLM in generating structured concepts that are short and distinctive. We refer to `https://github.com/sachit-menon/classify_by_description_release/blob/master/generate_descriptors.py` for the full script used in Menon & Vondrick (2023) to generate concepts.

## L  CLASS EMBEDDINGS WITH SUPPLEMENTARY INFORMATION

We provide an ablation experiment on using supplementary information of the class, taken from the original WordNet hierarchy. Specifically, we obtain two types of supplementary information: the broad parent of the object, and the more specific type of it. For example, the broad parent of the class "tench" is *animal*, and the specific type is *fish*. Similarly, the broad parent of the class "magpie" is *animal* and the specific type is *bird*. This supplementary information assists the textual encoder in

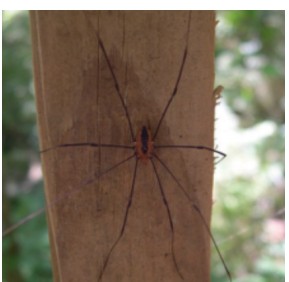

**prediction**: harvestman
*because...*
it shows spider in the genus, a species that grows on seeds and leaves

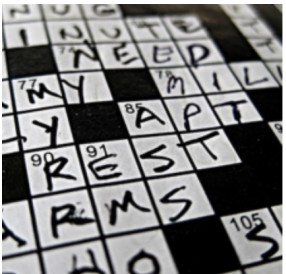

**prediction**: crossword puzzle
*because...*
it shows a vocabulary game with a word counter

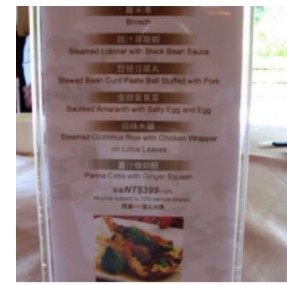

**prediction**: menu
*because...*
it shows items and prices, including price information on the product or service menu

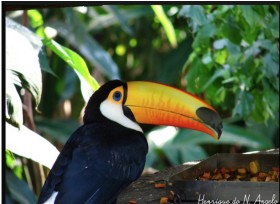

**prediction**: toucan
*because...*
it shows to be the orange, black or white color

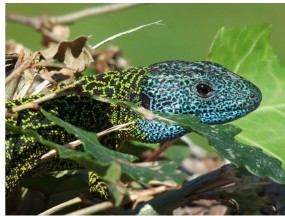

**prediction**: green lizard
*because...*
it shows lizard colors in the gardens of a reptilian plant, which is green

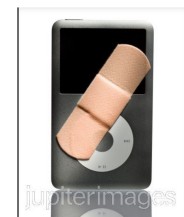

**prediction**: band aid
*because...*
it shows an image that is shown to a person using the product for curing wounds healing

Figure 9: Additional qualitative examples of our zero-shot NLEs on ImageNet for a ResNet-50 classifier.

better understanding the discriminative class names, particularly when it is unfamiliar with them, as the text encoder was trained on natural text that may not include these specific names.

It is important to note, however, that we refrain from using any of this supplementary information in our work. This is because the trained classifiers determine the parents implicitly on their own. Utilizing this information would facilitate more structured learning and could be considered as "unfair" or "biased". However, we still provide results on this for future research that may be relevant to our work. Results are presented in Table 12. As demonstrated, utilizing the broad parent does not significantly impact performance. In contrast, employing the type parent increases the Top-1 accuracy by approximately 1.5 points and the Top-5 accuracy by about 4 points. This improvement is attributed to the type parent (e.g., fish, bird) offering considerably more information to the text encoder about the object compared to the broad parent (e.g., animal).

Table 12: Ablation Studies on WordNet with a ResNet-50

|  | Top-1 | Top-5 |
|---|---|---|
| Only Class Names (Ours) | 33.28 | 66.20 |
| + WordNet (not used) | | |
| Broad Parent | 34.82 | 66.40 |
| Type Parent | **35.16** | **70.82** |

Finally, we also refrain from providing class descriptions as supplementary information, because these descriptions would leak to the explanations, which is the task we aim to achieve. Additionally, they would leak to other downstream tasks.

## M ADVANTAGES OVER CONTRASTIVE LANGUAGE-IMAGE PRETRAINING

An alternative option to our method is aligning a text encoder with visual features $f$ obtained from the frozen visual encoder $M_V$, via contrastive language-image pretraining. We provide two intuitive explanation of why our method of regressing class weights is better, efficient, and most importantly, faithful:

1. The output distribution of the visual classification model $M$ across all classes is given by $f.W$, from which the prediction is then made. Therefore, the output distribution highly depends on $W$, and not just the visual features $f$. By aligning a text encoder with the visual features $f$, we would be changing $W$, as they would now be the output of the newly aligned text encoder. Therefore, this approach is not faithful to the classification model, as it completely changes its whole output distribution. Our method on the other hand preserves the distribution by learning to regress the weights $W$.

2. Apart from the above fundamental problem, there is a significant technical challenge. Pretraining with contrastive learning requires a huge set of image-caption pairs (400 million at least) and a huge amount of computational resources, in order to reach the impressive performance of CLIP models. Performing this for each different classification model is not efficient, not ideal, and not desirable for users who wish to explain their classifiers. Our method on the other hand, can be trained on any moderate GPU (or even, a high-performing CPU), and takes around 10 seconds, which is considered negligible and can be applied to any visual classifier on-the-fly.

## N USING SOFT DROPOUT

For certain visual classifiers, we find that using learned soft dropout slightly harms performance. We hypothesize that this occurs when the weight embedding space of the visual classifier is already well-regularized, making additional regularization detrimental. In Table 13, we provide a complete list of our models, indicating whether or not they utilize the learned soft dropout. 20/38 models use it, while the remaining 18 does not. However, we note that the negative impact is minimal, and as such, retaining this technique across all models remains acceptable.

Table 13: Usage of Learned Soft Dropout

| Model | Soft Dropout | Model | Soft Dropout |
|---|---|---|---|
| ResNet50 | ✓ | ResNet101 | ✓ |
| ResNet50v2 | ✓ | ResNet101$_{v2}$ | ✓ |
| WideResNet50 | ✓ | WideResNet50$_{v2}$ | ✓ |
| WideResNet101 | ✓ | WideResNet101$_{v2}$ | ✓ |
| ResNeXt50$_{32x4d}$ | ✓ | ResNeXt50$_{32x4d, v2}$ | ✓ |
| ResNeXt101$_{64x4d}$ | ✓ | ResNeXt101$_{32x8d}$ | ✓ |
| ResNeXt101$_{32x8d, v2}$ | ✓ | DenseNet121 | ✓ |
| DenseNet161 | ✓ | EfficientNetv2-S | ✓ |
| EfficientNetv2-M | ✗ | ShuffleNetv2$_{x2.0}$ | ✗ |
| ConvNeXt-Tiny | ✗ | ConvNeXt-Small | ✗ |
| ConvNeXt-Base | ✓ | MobileNetv3-L | ✗ |
| ViT-B/32 | ✗ | ViT-B/16v2 | ✗ |
| ViT-L/32 | ✓ | ViT-L/16 | ✓ |
| Swin-T | ✗ | Swin-S | ✗ |
| ViT-B/16$_{pt}$ | ✓ | BeiT-B/16 | ✗ |
| BeiT-L/16 | ✗ | DINOv2-S | ✗ |
| DINOv2-B | ✗ | CvT-21 | ✗ |
| ConvNeXt-B$_{pt}$ | ✗ | ConvNeXtV2-B | ✗ |
| ConvNeXtV2-B$_{pt}$ | ✗ | ConvNeXtV2-B$_{pt, 384}$ | ✗ |

