# OpenReview forum: "Zero-Shot Natural Language Explanations"
_ICLR.cc/2025/Conference — ICLR 2025 Poster_

### Official Review · Reviewer_N3TX · 2024-10-20

**Soundness:** 3
**Presentation:** 2
**Contribution:** 3
**Rating:** 6
**Confidence:** 5

**Summary:**

The paper proposes a method to use natural language to explain the decisions made by an image classification model. To achieve this, they train a small MLP to map the text features to the feature space of the image classifier. Then, they use a pretrained language model and apply prefix-tuning, fine-tuning a small set of trainable parameters (added to the key-value pairs in the attention module). This enables the language model to generate explanations for a given image sample, using the image and the image classifier to extract its features.

**Strengths:**

- Provides a method for generating explanations for image classifier decisions using natural language.
- Does not require large-scale training data.
- Is inexpensive to train (only a small MLP and small prefixes in the attention mechanism).
- Includes good visualizations.
- Has strong paper motivation.
- Shows good performance on CNNs compared to supervised methods.

**Weaknesses:**

- Some parts of the paper are vague. For example:
  - The section discussing learnable prefixes set as key-value pairs in the attention modules. Could you please add one sentence to this part explaining how this works?
  - The last paragraph discussing zero-shot NLE generation (If you attach each of the $K$ tokens to the previously sampled top-K tokens, wouldn’t there be $K^2$ possible sentences instead of $K$?).
- It requires optimization for each image sample to produce an NLE.
- It requires loss calculation and backpropagation for each image sample to produce an NLE.
- Although the model performs well on unseen classes, it deteriorates performance on seen classes.
- LPIPS is not a suitable evaluation metric for this application because the NLE is not intended to generate captions for images (as T2I models require). Therefore, passing NLEs generated by this model to a T2I model to generate images does not make sense.
- No limitations are mentioned in the main text of the paper.

**Questions:**

- Regarding the last paragraph of section 3.2: If you attach each of the $K$ tokens to the previously sampled top-K tokens, how would there be $K$ possible sentences? Wouldn't it be $K^2$?
- In the first paragraph of section 4.1:
  - What is the number of classes in $\hat{W}$? (What is the dimension of $\hat{W}$?)
  - When training the MLP, which classes did you use? Only the 900 classes, or all of them?
- As a metric to evaluate your method, the cosine similarity between the text embeddings of the predicted explanation and the ground truth explanation (from ImageNet-X) using the BERT model is interesting and missing. Have you tried this?
- Do you have an intuition about why your model performs worse than other supervised models on ViT models (even though it performs better on CNNs)? Do you have any experimentations to support your intuition?

---

> ### Author Response · Authors · 2024-11-19
> **Response to reviewer N3TX (Part 1)**
>
> Dear Reviewer,
>
> We thank you for your time in reviewing our paper and for your valuable feedback, and for the strengths you provided. We address all your concerns below.
>
> > The section discussing learnable prefixes set as key-value pairs in the attention modules. Could you please add one sentence to this part explaining how this works?
>
> We use learnable prefixes, attached to the embedding space of the language model, to steer the language model to generate explanations of the visual features. To do so, the learnable prefixes are set as initial key-value tokens preceding the explanation generation, so each generated word in the explanation can attend to these prefixes. If we have $Y$ currently generated words, and we use 5 prefixes, then at each timestep, the number of keys and value tokens in the attention mechanism will always be ($5 + Y$), while the number of query tokens will be $Y$. Therefore, the $Y$ query tokens will always attend to the $Y$ key-value tokens *and* the $5$ prefixes (so $5 + Y$ tokens to attend to). We will clarify these aspects in Section 3.2 of the revised manuscript.
>
> > The last paragraph discussing zero-shot NLE generation (If you attach each of the K tokens to the previously sampled top-K tokens, wouldn’t there be K^2 possible sentences instead of K?).
>
> We apologize for this confusion. At a given timestep, the model samples $K$ tokens from the Pretrained Language Model (PLM) distribution $o_l$, which act as $K$ continuation tokens of the currently generated explanation. These $K$ potential sentences are then passed to the text encoder, followed by the MLP to produce $K$ similarity scores with the visual feature vector $f$. These scores, normalized with softmax, form a target distribution to train against $o_l$ with cross-entropy, and the prefixes of the PLM are updated with backpropogation. We then run the PLM again with the updated prefixes and we sample from the output distribution the most likely token (token with the highest logit score). Namely, per timestep, $K$ token are initially produced to update the prefixes but one token is finally sampled (after the prefixes are updated) as the continuation of the explanation. We run the above process for $L$ timesteps. We set $L$ as the maximum sequence length we want for the explanation, or until the $<.>$ token is reached. After these $L$ timesteps, one iteration will be concluded, and one explanation will be generated. We train for 20 iterations, producing 20 explanations from which we select the final explanation as the one that maximizes the CLIP-score with the image. The text in Section 3.2 and Figure 4 will be properly updated to clarify these aspects.
>
> > It requires optimization for each image sample to produce an NLE. It requires loss calculation and backpropagation for each image sample to produce an NLE.
>
> Indeed, unlike supervised NLE models which require a single forward pass at inference, our approach is more time-consuming. However, as mentioned in L235 of the manuscript, our approach allows for flexibility in the generated text, operates in an open-set environment, and can be adapted to any example or classifier on the fly. We believe that these factors are more important and unique in NLEs, thereby outbalancing the increased explanation generation time.
>
> > Although the model performs well on unseen classes, it deteriorates performance on seen classes.
>
> Indeed, the baseline ICIS wins on the seen classes. This is due to the two dropout regularizations that we add. With the two dropout mechanisms, the model's capacity is effectively reduced, which lowers its ability to fit the training data perfectly, resulting in lower training accuracy. However, 1) the dropout mechanisms also lead to significant improvements on the unseen data, and 2) for most models, the accuracy drop with respect to the baseline ICIS is only around 5%. Considering these two factors, we prioritized gains in unseen classes over those in seen ones.
>
> > No limitations are mentioned in the main text of the paper.
>
> Due to space limitations, we were unable to fit the limitations section in the main paper, so we deferred it to the first section of the supplementary material (section A, page 15) and also referenced that section in the main manuscript (in the Conclusion section).
>
> > What is the dimension of $\hat{W}$?
>
> Implementation details such as the dimensions of the MLP ($\hat{W}$) are provided in the Section F in the supplementary material. We kindly refer you to page 17, first paragraph of section F in the supplementary material. The MLP maps the text encoder features to class embeddings of the classifier. Therefore, the input dimension to the MLP will always be the same as the dimension of the text encoder (in our case, 768), and the output dimension of the MLP will be the same as the dimension of the class embeddings of the visual classifier (e.g., 2048 for ResNets, 768 for ViT-B).

---

> > ### Comment · Reviewer_N3TX · 2024-11-21
> >
> > > Indeed, unlike supervised NLE models which require a single forward pass at inference, our approach is more time-consuming. However, as mentioned in L235 of the manuscript, our approach allows for flexibility in the generated text, operates in an open-set environment, and can be adapted to any example or classifier on the fly. We believe that these factors are more important and unique in NLEs, thereby outbalancing the increased explanation generation time.
> >
> > Inference time is crucial since it occurs during every run. Additionally, this approach cannot batch operations during inference, which limits its ability to accelerate NLE generation.
> >
> > >Indeed, the baseline ICIS wins on the seen classes. This is due to the two dropout regularizations that we add. With the two dropout mechanisms, the model's capacity is effectively reduced, which lowers its ability to fit the training data perfectly, resulting in lower training accuracy. However, 1) the dropout mechanisms also lead to significant improvements on the unseen data, and 2) for most models, the accuracy drop with respect to the baseline ICIS is only around 5%. Considering these two factors, we prioritized gains in unseen classes over those in seen ones.
> >
> > So, are you saying that other methods have overfitted to the training data, performing well on it but not as well as yours on the validation set?

---

> ### Author Response · Authors · 2024-11-19
> **Response to Reviewer N3TX (Part 2)**
>
> > When training the MLP, which classes did you use? Only the 900 classes, or all of them?
>
> This depends on the application. For zero-shot image classification, we only train on the 900 classes and test on the 100 unseen classes. For generating NLEs, we train on the full set of classes (1,000 for ImageNet), as mentioned in L328.
>
> > As a metric to evaluate your method, the cosine similarity between the text embeddings of the predicted explanation and the ground truth explanation (from ImageNet-X) using the BERT model is interesting and missing. Have you tried this?
>
> Following your suggestion, we assess our zero-shot NLEs against supervised NLEs with the BERT-Score. The results presented below show that our zero-shot explanations are outperformed by the supervised ones. This is expected as the supervised NLEs are explicitly trained to approximate the ground-truth ones. Moreover, unlike the other evaluation metrics we used, the BERT-Score is completely agnostic of the image, which is the reason why we avoided it in the first place.
>
> |Model|Bert-Score|
> |-|-|
> |nlxgpt_FT|0.860|
> |nlxgpt_noFT|**0.865**|
> |uninlx_FT|0.862|
> |uninlx_noFT|0.862|
> |**Ours**||
> |ResNet50|0.716|
> |BeiT-L|0.714|
> |ConvNeXtv2-B (pt)|0.713|
> |DenseNet161|0.715|
> |DiNOv2-Base|0.714|
> |ResNeXt50 (32x4d)|0.716|
> |ViT-B/16 (pt)|0.717|
> |WideResNet101|0.717|
> |WideResNet50|0.718|
>
> > Do you have an intuition about why your model performs worse than other supervised models on ViT models (even though it performs better on CNNs)?
>
> We analyzed all our models across their original performance, their performance on zero-shot accuracy with our method, and their weight embedding size. We find the following: (i) better-performing models have more discriminative features that are more distributed across space than less-performing models, making it more difficult to regress the weights $W$; (ii) ViTs have typically smaller dimensions of $W$ (e.g., 768 or 1024) which leads to a smaller number of constraints when we are learning the regression problem to find $W$; (iii) The $W$ of ViTs have a lack of inductive bias and local structure as opposed to convolutional models leading to more difficulty in regressing them. We will add this discussion to the manuscript.
>
> > LPIPS is not a suitable evaluation metric for this application because the NLE is not intended to generate captions for images (as T2I models require). Therefore, passing NLEs generated by this model to a T2I model to generate images does not make sense.
>
> This is true. However, the explanation can be re-formulated as a prompt to the text-to-image model. For example, for the NLE in Figure 6: “because it shows serpentine rings on the snake skin,” we can re-formulate the prompts as: “a realistic photo of serpentine rings on the snake skin.” Therefore, the image that will be synthesized by the text-to-image model will present a snake with serpentine rings. If the textual features described by the NLE are correct and faithful to the model, the semantic LPIPS metric between the synthesized image and the original image (which generated the NLE) will have a better score. When the NLE is correct and faithful, the synthesized image will also activate the visual features of the classifier, causing them to have a high cosine similarity score with those of the original image.

---

> > ### Comment · Reviewer_N3TX · 2024-11-21
> >
> > >Following your suggestion, we assess our zero-shot NLEs against supervised NLEs with the BERT-Score. The results presented below show that our zero-shot explanations are outperformed by the supervised ones. This is expected as the supervised NLEs are explicitly trained to approximate the ground-truth ones. Moreover, unlike the other evaluation metrics we used, the BERT-Score is completely agnostic of the image, which is the reason why we avoided it in the first place.
> >
> > I think this suggests that the other methods perform better than yours. As you know, BERT-score evaluates the similarity between two texts in the embedding space. If your model's explanations were closer to the ground truth explanations (human annotations), your score would be higher — but that’s not the case.
> >
> > >We analyzed all our models across their original performance, their performance on zero-shot accuracy with our method, and their weight embedding size. We find the following: (i) better-performing models have more discriminative features that are more distributed across space than less-performing models, making it more difficult to regress the weights $W$; (ii) ViTs have typically smaller dimensions of $W$ (e.g., 768 or 1024) which leads to a smaller number of constraints when we are learning the regression problem to find $W$; (iii) The $W$ of ViTs have a lack of inductive bias and local structure as opposed to convolutional models leading to more difficulty in regressing them. We will add this discussion to the manuscript.
> >
> > This suggests that your method may not be well-suited for high-performing state-of-the-art models, which are mostly ViT-based.
> >
> > >This is true. However, the explanation can be re-formulated as a prompt to the text-to-image model. For example, for the NLE in Figure 6: “because it shows serpentine rings on the snake skin,” we can re-formulate the prompts as: “a realistic photo of serpentine rings on the snake skin.” Therefore, the image that will be synthesized by the text-to-image model will present a snake with serpentine rings. If the textual features described by the NLE are correct and faithful to the model, the semantic LPIPS metric between the synthesized image and the original image (which generated the NLE) will have a better score. When the NLE is correct and faithful, the synthesized image will also activate the visual features of the classifier, causing them to have a high cosine similarity score with those of the original image.
> >
> > The issue is that image generation and NLE are fundamentally different tasks. The captions needed for image generation and the explanations produced by NLE methods serve distinct purposes and should differ accordingly. If an explanation generated by an NLE method closely resembles a caption designed for image generation, it indicates a potential problem. In other words, prompts for image generation are unrelated to the objectives of NLE.

---

> ### Author Response · Authors · 2024-11-22
> **Response to reviewer (part 1)**
>
> > Inference time is crucial since it occurs during every run.
>
> Firstly, as outlined in our initial response, our approach prioritizes key factors such as flexibility, open-setting applicability (generating words outside the dataset corpus), and the ability to generate explanations adaptable to any classifier on-the-fly. Achieving these objectives necessitates certain trade-offs, in this case, time efficiency. Additionally, the time efficiency of our approach is quite reasonable, averaging approximately 15-20 seconds per sample. Secondly, we kindly draw your attention to the fact that this limitation was explicitly acknowledged in Section A of the appendix (page 15). While we understand your expectations, it is important to note that all research works inherently come with their own set of limitations, which we have addressed thoroughly in Section A of the appendix.
>
> > So, are you saying that other methods have overfitted to the training data, performing well on it but not as well as yours on the validation set?
>
> We did not make this claim. Rather, we stated that our method overfits *less* to the training data compared to other methods. We mentioned in our response: *"it lowers its ability to fit the training data"*. This does not imply that other methods overfit the training data.
>
> > The issue is that image generation and NLE are fundamentally different tasks. The captions needed for image generation and the explanations produced by NLE methods serve distinct purposes and should differ accordingly.
>
> The NLEs is a part of this form: “<prediction> because it shows <NLE>”.  The text-to-image model prompt will then be in this format: “a realistic image showing <NLE>”.  The NLE consists of textual descriptors, concepts, or features that describe the prediction. This is evident in our qualitative examples in Figure 6 (in the main paper) and Figure 9 (in supp.). They describe descriptors or features in text such as (“cart buckets”, “tools at a farm”), (“honey bees”, “farm bee factory”), (“serpentine ring on snake skin”), (“head as a green lizard”), (“jellyfish glow”), (“vocabulary game”, “word counter”), (“price information”, “service menu”)....etc. These descriptors/features are formed in a natural language expression. Therefore, the synthesized image will depict an image incorporating these features. Here are some examples of prompts we use for the NLEs in Figure 6 and 9:
>
> <a realistic image showing tractor cart buckets and other tools in the field at a farm > \
> <a realistic image showing honey bees in the food bee factory at a farm for Honeybee Foods > \
> <a realistic image showing serpentine ring on the snake skin> \
> <a realistic image showing jellyfish glow in a tube on the screen > \
> <a realistic image showing a vocabulary game with a word counter > \
> <a realistic image showing items and prices, including price information on the product or service menu >
>
>
> In conclusion, it is straight forward to formulate the explanation as a text prompt that the text-to-image model can understand.

---

> ### Author Response · Authors · 2024-11-22
> **Response to reviewer (part 2)**
>
> > BERT-Score: I think this suggests that the other methods perform better than yours.
>
> We kindly ask you to note two important points:
>
> - Supervised methods are naturally expected to perform better, as they are explicitly trained on annotated data. Achieving performance where zero-shot methods surpass supervised ones is highly challenging. The results for supervised methods are included to provide an *upper bound* for comparison with zero-shot methods.
>
> - We would like to reiterate that the BERT-Score is entirely agnostic to the image, as it does not see any visual information and, therefore, cannot assess the alignment between the image and the explanation. It is well-recognized in the field of zero-shot vision-language methods that relying on a metric that considers only one modality (i.e.., only text or only vision) is suboptimal for evaluation.
>
> > This suggests that your method may not be well-suited for high-performing state-of-the-art models, which are mostly ViT-based.
>
> That is not necessarily true. To show this, we conducted experiments on two other tasks. The first task is zero-shot transfer. That means we take the MLP trained to synthesize the 1,000 ImageNet class weights and test it on synthesizing class weights for other datasets. We choose Places365 dataset (specializing in scene classification) and DTD dataset (specializing in texture type classification). These are challenging datasets because even the powerful CLIP model trained on 400 million image-text pairs, achieves as low as 41.7% top-1 zero-shot accuracy on DTD and 37.37% on Places365. We report the Test@1 and Test@5 Generalized Zero-Shot Accuracy, on the **full** validation set of those datasets. In this case, the generalized Zero-Shot accuracy on the test data is the same as the Zero-Shot Setting (since we validate on all unseen classes of the Places365/DTD dataset). We kindly remind you that the objective of this experiment is not to outperform CLIP, as our method is not comparable with CLIP since it is trained without *any* image data (text-only training) and using 1,000 samples only (400,000$\times$ less data than CLIP). Instead, the objective of this experiment is to show that high-performing state-of-the-art models based on large-scale pretraining achieve much better results on this task for both these two challenging datasets.
>
> ||||||
> |-|-|-|-|-|
> |**Model**|**Places365**||**DTD**||
> ||Test@1|Test@5|Test@1|Test@5|
> |**ResNet50**|10.76|30.92|14.36|32.29|
> |**WideResnet101**|13.02|33.89|12.77|31.97|
> |**ViT-B/32**|13.89|34.41|15.74|33.24|
> |**ViT-L/16**|14.17|33.91|15.53| 32.39|
> |**ConvNeXt-Base**|15.03|35.92|18.46|34.79|
> |**DiNOv2-Base**|16.66|40.88|17.66|40.05|
> |**ViT-B/16 (pt)**|18.46|44.53|15.85|33.30|
> |**ConvNeXtv2-B (pt-384)** |18.48|42.44|19.20|35.96|
> |**BeiT-L/16**|**19.18**|**43.59**|**20.27**|**37.13**|
>
> The second task we use to address your concern is zero-shot image captioning on the COCO dataset. This is a dataset deferring in distribution from ImageNet in terms of context and object composition in images. It is worth noting that the ground-truth annotated captions are not considered as explanations, because they are descriptions of images written by humans. However, they serve as good approximations for explanations. We report results on the common Karpathy test split, and compare our method with two other current state-of-the-art zero-shot baselines: ZeroCap and ConZic. We use the common metrics for evaluation: BLEU-4 (B4), METEOR (M), CiDEr (C) and SPICE (S). During the past few days, we were able to conduct experiments for 6 models. The results are shown below and they show that high-performing state-of-the-art models based on large-scale pretraining, achieve better results on the most important metrics in the captioning task (SPICE , CiDEr). In fact, our method with BeiT-L sets a new state-of-the-art on zero-shot image captioning.
>
> | Method|B4|M|C|S|
> |-|:-:|:-:|:-:|:-:|
> |ZeroCap|**2.6**|11.5|14.6| 5.5|
> |ConZIC|1.3|11.5|12.8|5.2|
> |**Ours**|||||
> |ResNet50|1.7|11.2|14.9|6.6|
> |ResNet101|1.8|11.3|15.5|6.6|
> |DINOv2-B|1.8|11.6|16.2|7.1|
> |ConvNeXtV2-B (pt-384)|2.0|11.5|16.9|7.2|
> |ViT-B/16 (pt)|1.7|**11.9**|17.4|7.4|
> |BEiT-L/16|1.8|11.7|**17.6**|**7.6**|

---

> > ### Comment · Reviewer_N3TX · 2024-11-25
> >
> > Thank you for your responses. I will keep my score as it is.

---

### Official Review · Reviewer_Y6Z6 · 2024-10-21

**Soundness:** 2
**Presentation:** 3
**Contribution:** 2
**Rating:** 6
**Confidence:** 3

**Summary:**

This paper presents a zero-shot natural language explanation approach.
The proposed method involves two sequential steps.
For the first step, the authors propose to align the text encoder with that of the classifier from a visual encoder.
Subsequently, an LLM is then equipped with a prefix that is trained by the pseudo label learned from the former stage.
The experiments are primarily focused on the NLE task.
When compared with several baselines, the proposed method achieves improved performance on diverse vision backbones.

**Strengths:**

- The studied problem, zero-shot natural language explanation, is quite practical and useful for many researchers.
- The writing of this paper is good, and most parts of it are easy to follow.
- The authors utilize a variety of vision encoders to validate the generalization capability of the proposed method.

**Weaknesses:**

- My biggest concern is why we simply remove the first stage and use the CLIP model for the alignment.
To me, the authors try to align the output from a text encoder with that of a vision encoder classifier.
This process is utilized to offer a pseudo label for the subsequent LLM prefix-tuning.
I feel like we can directly use a pre-trained CLIP as a replacement which can generalize well to open settings.
- Following the first concern, how did the authors address out-of-distribution classes?
It seems the classifiers are trained on 1,000 ImageNet classes.
What if there are more fine-grained classes?
- It is controversial to call this method a ``classifier-agnostic`` method.
  - The authors did use a classifier in the first training stage.
  - The MLE is evaluated on the ImageNet domain.
- There is only 1 baseline compared in Table 1.
Additionally, the improvement over this method is not significant.

**Questions:**

One more question other than the aforementioned weaknesses:

How about the model performance using a tiny vision large language model (VLLM)?

---

> ### Author Response · Authors · 2024-11-19
> **Response to reviewer Y6Z6 (Part 1)**
>
> Dear Reviewer,
>
> We thank you for your time and effort in reviewing our paper, and for the feedback and strengths you provided for our work. We will clarify all your concerns below:
>
> > My biggest concern is why we simply remove the first stage and use the CLIP model for the alignment. I feel like we can directly use a pre-trained CLIP as a replacement
>
> The goal of the research is to produce a faithful natural language explanation for *any* visual classifier. Assume that you have a visual classifier trained on a dataset in a supervised manner. This classifier is not the CLIP visual encoder. The feature vectors produced by this classifier are not aligned with a textual model, so we cannot access or query this visual classifier via text. If we were to replace the proposed alignment with using directly the CLIP model as the reviewer suggests, then we would generate explanations for the CLIP visual encoder rather than the classifier.
>
> > Following the first concern, how did the authors address out-of-distribution classes?
>
> In this work, we do not deal with the OOD problem. We are only concerned with creating explanations for the trained classifier. The method is not explicitly designed to address explanations for OOD samples, because the classifier itself may fail on the OOD samples (or does not understand them). However, please note that our method is suited to datasets that have some shared properties with ImageNet, such as Places365, specializing in scene classification, and DTD, specializing in texture type classification. To show this, we report results on zero-shot transfer. That means we take the MLP trained to synthesize ImageNet class weights and test it on synthesizing class weights for Places365 and DTD. We report the Test@1 and Test@5 Generalized Zero-Shot Accuracy, on the **full** validation set of those datasets. In this case, the generalized Zero-Shot accuracy on the test data is the same as the Zero-Shot Setting (since we validate on all unseen classes of the other dataset). Note that these datasets (Places365, DTD) are challenging, and even the powerful CLIP model trained on 400 million image-text pairs, achieves as low as 41.7% top-1 zero-shot accuracy on DTD and 37.37% on Places365. On the other hand, our method is trained without any image data (text-only training) and using 1000 samples only (400,000$\times$ less data). Our results are below for several classifiers:
>
> ||||||
> |-|-|-|-|-|
> |**Model**|**Places365**||**DTD**||
> ||Test@1|Test@5|Test@1|Test@5|
> |**ResNet50**|10.76|30.92|14.36|32.29|
> |**WideResnet101**|13.02|33.89|12.77|31.97|
> |**ViT-B/32**|13.89|34.41|15.74|33.24|
> |**ViT-L/16**|14.17|33.91|15.53| 32.39|
> |**ConvNeXt-Base**|15.03|35.92|18.46|34.79|
> |**DiNOv2-Base**|16.66|40.88|17.66|40.05|
> |**ViT-B/16 (pt)**|18.46|44.53|15.85|33.30|
> |**ConvNeXtv2-B (pt-384)** |18.48|42.44|19.20|35.96|
> |**BeiT-L/16**|19.18|43.59|20.27|37.13|
>
> We observe a similar trend as in the literature of OOD. Models based on large-scale pre-training (especially Transformers) achieve better results in OOD performance. This trend is consistent across the two datasets we evaluate on.
>
> To address your concern even better, we ran experiments on the COCO dataset. This is a dataset deferring in distribution from ImageNet in terms of context and object composition in images. We used zero-shot image captioning as the task. It is worth noting that the ground-truth annotated captions are not considered as explanations, because they are descriptions of images written by humans. However, they serve as good approximations for explanations. We report results on the common Karpathy test split, and compare our method with two other current state-of-the-art zero-shot baselines: ZeroCap and ConZic. We use the common metrics for evaluation: BLEU-4 (B4), METEOR (M), CiDEr (C) and SPICE (S). During the past few days, we were able to conduct experiments for 6 models. The results are shown below and they show that our method with BeiT-L, outperforms the baselines on the most important metrics in the captioning task (SPICE , CiDEr) and sets a new state-of-the-art on zero-shot image captioning.
>
> | Method|B4|M|C|S|
> |-|:-:|:-:|:-:|:-:|
> |ZeroCap|**2.6**|11.5|14.6| 5.5|
> |ConZIC|1.3|11.5|12.8|5.2|
> |**Ours**|||||
> |ResNet50|1.7|11.2|14.9|6.6|
> |ResNet101|1.8|11.3|15.5|6.6|
> |DINOv2-B|1.8|11.6|16.2|7.1|
> |ConvNeXtV2-B (pt-384)|2.0|11.5|16.9|7.2|
> |ViT-B/16 (pt)|1.7|**11.9**|17.4|7.4|
> |BEiT-L/16|1.8|11.7|**17.6**|**7.6**|

---

> > ### Author Response · Authors · 2024-11-19
> > **Response to reviewer Y6Z6 (Part 2)**
> >
> > > It is controversial to call this method a classifier-agnostic method
> >
> > You are correct in that this is not a classifier-agnostic method, as it requires to train an MLP for each classifier separately. However, we mention in L186 that the time for training the MLP is roughly 10 seconds and is therefore considered negligible and thus can be applied instantly to any classifier on the fly. That is why we framed our method as “classifier-agnostic”. However, we acknowledge that, indeed, this does not make our method classifier-agnostic, and we will change this in the manuscript and in the title of our paper.
> >
> > >There is only 1 baseline compared in Table 1. Additionally, the improvement over this method is not significant.
> >
> > To the best of our knowledge, ICIS baseline is the state-of-the-art in text-only training for zero-shot classification. As a result, comparing our results to previous methods (before ICIS) appears redundant. Kindly also note that the improvement is significant for most models. The most important metric is the generalized zero-shot accuracy on the test set (unseen classes). For example, the improvement in Top-1 Accuracy is as high as 21% (for ResNeXt50). The improvement in the Zero-Shot Setting (rather than the Generalized Zero-Shot Setting) is indeed not significant. However, as mentioned in L320, this setting is not realistic, as this setting only considers the unseen classes in the pool of available classes (it totally removes the seen classes from the pool of classes to classify from).

---

> > ### Comment · Reviewer_Y6Z6 · 2024-11-21
> > **Clarification about the Relationship with CLIP**
> >
> > I appreciate the OoD results provided by the authors though the performance is less satisfactory.
> >
> > I'd like to clarify the comment that links CLIP with the key contribution of this work.
> > To me, it seems like using CLIP pre-training for vision encoders will generally deliver better results, especially OoD results, than ImageNet pre-training ones.
> > Besides, we can easily build classifier weights with the output of the CLIP text encoder.
> > I understand that some of these visual encoders are not accompanied by CLIP pre-training.
> >
> > The authors can address this comment by either some experiments pre-training with CLIP or intuitive explanations of why the current implementation is superior to CLIP ones.

---

> ### Author Response · Authors · 2024-11-21
> **Response about the relationship with CLIP**
>
> We thank you for clarifying this concern. We assume that by "pre-training with CLIP", you mean aligning a text encoder with visual features $f$ obtained from the frozen visual encoder $M_V$, via contrastive language-image pretraining. We provide two intuitive explanation of why our method of regressing class weights is better, efficient, and most importantly, faithful:
>
> - The output distribution of the visual classification model $M$ across all classes is given by $f.W$, from which the prediction is then made. Therefore, the output distribution highly depends on $W$, and not just the visual features $f$. By aligning a text encoder with the visual features $f$, we would be changing $W$, as they would now be the output of the newly aligned text encoder. Therefore, this approach is not faithful to the classification model, as it completely changes its whole output distribution. Our method on the other hand preserves the distribution by learning to regress the weights $W$.
>
> - Apart from the above fundamental problem, there is a significant technical challenge. Pretraining with contrastive learning requires a huge set of image-caption pairs (400 million at least) and a huge amount of computational resources, in order to reach the impressive performance of CLIP models. Performing this for each different classification model is not efficient, not ideal, and not desirable for users who wish to explain their classifiers. Our method on the other hand, can be trained on any moderate GPU (or even, a high-performing CPU), and takes around 10 seconds, which is considered negligible and can be applied to any visual classifier on-the-fly.
>
> We hope that we have addressed your concern.

---

> > ### Comment · Reviewer_Y6Z6 · 2024-11-22
> > **Concerns addressed**
> >
> > The second response addresses my concerns. I'd like to raise my score.

---

### Official Review · Reviewer_9tNT · 2024-10-25

**Soundness:** 3
**Presentation:** 3
**Contribution:** 4
**Rating:** 8
**Confidence:** 3

**Summary:**

The paper proposes a new method for zero-shot natural language explanation (NLE) for classification. The method trains an MLP that maps the encoding space of an LLM text encoder to the space of class weights of arbitrary classifiers. Consequently, the method works on arbitrary classifiers. It produces NLE by iteratively adapting a prefix to the image classifier’s feature representation, and the prefix serves as a condition for generating text descriptions. In this way, the generated text is faithful, in contrast to the case of models trained on human description data specifically. It achieves the state-of-the-art performance on zero-shot NLE, and is also on par to the SOTA methods in zero-shot classification and concept discovery.

**Strengths:**

1. The method proposed by the paper is novel, taking a completely different perspective from that of the previous works in the field of NLE.
2. The method is powerful, in that it achieves the state-of-the-art performance on zero-shot NLE, and is also on par to the SOTA methods in zero-shot classification and concept discovery.
3. The method also demonstrates and utilises the possibility of matching the representation space of LLM to arbitrary vision classifier (or any models whose outputs are texts), which gives deep insights into this mechanism and can be inspiring to other fields as well.
4. Overall, the presentation of the paper is very clear.

**Weaknesses:**

1. Section 3.2 (as well as Section F) is somewhat confusing to me. From my understanding, the generation of NLE is the following process: 1. there are K sentences starting with some initial prompts like “this is a”; 2. a random prefix is present to generate top-K tokens; 3. the top-K tokens are concatenated to the K sentences to get K continued sentences; 4. the representation of these sentences are compared to the image representation, and the prefix is updated to reflect the similarities, so that next step it can generate better tokens that make the sentences aligned to the image; 5. the whole process continues so that the sentences become longer. I still have the following doubts (I will update my score and confidence once I understand them better; I also think Figure 4 should be more clear on this whole process):
    1. At the first time step, the sentences will become “this is a [object]”. How to make sure the top-K tokens actually contain the correct object, so that the continuation does not follow a completely wrong object?
    2. How is it possible for the generated sentence to automatically have “because” in it, so that the NLE is generated to be “this is a [object], because…”
    3. In Section F, I am a bit confused between “iterations” and “time steps”
    4. In Section F, I am not sure why the CLIP-score is used to choose the sentences. Shouldn’t it be that the sentence is chosen based on its similarity to the image representation? And is it fair to use CLIP-score here when comparing to other methods?
2. In the NLE comparison results, it seems that with the “cosine” metric the proposed model seems to be systematically inferior than the best models. The author might want to analyse the cause and give one explanation.

**Questions:**

1. The author might consider improving the presentation in section 3.2 by making the process more clear.
2. The author might want to analyse the cause and give one explanation for the inferior "cosine" metric in NLE results.
3. I wonder if this method can be applied to directly improve the robustness of image classifiers.

---

> ### Author Response · Authors · 2024-11-19
> **Response to reviewer 9tNT**
>
> Dear reviewer,
>
> Thank you very much for your time in reviewing our paper and for your constructive and valuable feedback. Thank you also for the strengths you provided for our paper and for your positive score. We address your concerns below.
>
> > At the first time step, the sentences will become “this is a [object]”. How to make sure the top-K tokens actually contain the correct object, so that the continuation does not follow a completely wrong object?
>
> > In Section F, I am a bit confused between “iterations” and “time steps”
>
> We address both these concerns here. We apologize for this confusion. At a given timestep, the model samples $K$ tokens from the Pretrained Language Model (PLM) distribution $o_l$, which act as $K$ continuation tokens of the currently generated explanation. These $K$ potential sentences are then passed to the text encoder, followed by the MLP to produce $K$ similarity scores with the visual feature vector $f$. These scores, normalized with softmax, form a target distribution to train against $o_l$ with cross-entropy, and the prefixes of the PLM are updated with backpropogation. We then run the PLM again with the updated prefixes and we sample from the output distribution the most likely token (token with the highest logit score). Namely, per timestep, $K$ token are initially produced to update the prefixes but one token is finally sampled (after the prefixes are updated) as the continuation of the explanation. We run the above process for $L$ timesteps. We set $L$ as the maximum sequence length we want for the explanation, or until the $<.>$ token is reached. After these $L$ timesteps, one iteration will be concluded, and one explanation will be generated. We are generating a whole sentence at each iteration and not a word per iteration. Therefore, each iteration generates one complete explanation. We train for 20 iterations, producing 20 explanations from which we select the final explanation as the one that maximizes the CLIP-score with the image. As a result, while non-relevant tokens may be sampled at the first iteration resulting in a non-relevant sentence, these tokens will have their scores reduced and will not likely be sampled in the next iteration. In our experiments, we observed that relevant tokens generally start appearing from the second iteration already. The text in Section 3.2 and Figure 4 will be properly updated to clarify these aspects.
>
> > How is it possible for the generated sentence to automatically have “because” in it, so that the NLE is generated to be “this is a [object], because…”
>
> The word “because” is added in the paper from us to signify the explanation part; it is not systematically produced by the model. To avoid any confusion, we will replace the word “because” by “generated explanation:” below the “Prediction:” in Figure 6 of the revised paper.
>
> > In Section F, I am not sure why the CLIP-score is used to choose the sentences. Shouldn’t it be that the sentence is chosen based on its similarity to the image representation? And is it fair to use CLIP-score here when comparing to other methods?
>
> The similarity to the image representations is used to construct the target distribution to train the prefixes in the PLM, producing an explanation at each iteration. We choose the CLIP-Score as the metric to select the best explanation because it is not biased to the process (and metric) used to create those explanations.
>
> > In the NLE comparison results, it seems that with the “cosine” metric the proposed model seems to be systematically inferior than the best models. The author might want to analyse the cause and give one explanation.
>
> Given two feature vectors representing two images, the cosine similarity only captures the geometric alignment of these vectors in the Euclidian space. To set an example, consider two vectors that are scalar multiples of each other (i.e., aligned in the Euclidian space): $v_1$ = $[1,1,0]$, $v_2$ = $[2,2,0]$; namely, $v_2$ is just $2v_1$. The cosine similarity between those two vectors is 1, indicating perfect alignment. Therefore, cosine similarity only measures the direction of vectors (or their angle) and does not capture semantics well. On the other hand, the CLIP-Score and the LPIPS metrics capture more semantically relevant alignment. LPIPS is also a learned metric trained with human input and has been shown to correlate very well with human judgment. We will add this discussion to the manuscript.

---

> > ### Comment · Reviewer_9tNT · 2024-11-21
> > **Response to authors**
> >
> > My concerns have been addressed, which boost my confidence in my evaluation. I personally think that the authors might refine section 3.2 a bit more to reflect the relation between "time steps" and "iterations", possibly by giving a more general overview first. I also think more details in section F should be integrated into section 3.2.

---

### Official Review · Reviewer_zJsF · 2024-10-28

**Soundness:** 3
**Presentation:** 3
**Contribution:** 3
**Rating:** 6
**Confidence:** 3

**Summary:**

The paper primarily explores generative natural language explanations for vision models. Alternatively, the paper considers learning a simple MLP to retrieve text most similar to the visual embedding of a given vision model. They claim to be the first work to this in a zero-shot manner. The authors conduct extensive ablations across various models and demonstrate strong results.

**Strengths:**

1. The proposed method is simple and intuitive, I liked the motivation behind the work.
2. The paper is well written and easy to follow, this makes it easy to understand what the authors have done and the current State of The Art.
3. The authors experiment on a wide variety of models.
4. The authors are able to extend "Visual Classification via Description from Large Language Models", in a practical manner to a wide variety of models.

**Weaknesses:**

1. Based on Table 1, the simple method seems to work less well for bigger models such as ViT and DEIT in the generalized zero-shot setting. There is a lack of discussion of the results, a little discussion about Table 1 would be helpful.
2. Since this is a zero-shot method, it would have been cool to see how does it work on other popular yet specialized datasets like Flowers / Aircraft / CUB etc. Although ImageNet results are good having more datasets would have given more insights.

**Questions:**

1. Did the authors ablate the prompt template? CLIP can be sensitive to prompt templates, was a similar trend seen for the learnt MLP?
2. Were there any biases in the generated explanations? Like bias towards color or specific features?

---

> ### Author Response · Authors · 2024-11-18
> **Response to reviewer zJsF**
>
> Dear Reviewer,
>
> We thank you for your time in reviewing our paper and for your constructive and valuable feedback, and for the strengths you provided. We address all your concerns below. All the discussions and analysis below will be added to our manuscript.
>
> > Since this is a zero-shot method, it would have been cool to see how does it work on other popular yet specialized datasets like Flowers / Aircraft / CUB etc.
>
> Our work is not well-suited for fine-grained datasets (such as specific types of flowers, aircrafts...etc). This is a limitation of our work which we will acknowledge. This is because: 1) the text encoder (referred to as $M_T$ in our work) does not understand fine-grained class names, as it was trained on general text. Let's take two classes from the Flowers102 dataset as an example: “poinsettia” and “monkshood”. The text encoder is not even aware that the two classes are a type of flower, as their similarity from the text encoder is 0.2. Unlike ImageNet, no other known classes can assist the unknown ones (as we describe in L190). Therefore, no meaningful embeddings can be learned by the MLP. 2) The number of classes (which are the number of data samples in our method) for these fine-grained datasets are typically very small (102 classes for Flowers102, 100 classes for FGVC Aircraft), which makes the regressing problem more challenging.  3) For NLE experiments, we do not have any annotations for these datasets, and therefore we cannot conduct evaluations. 4) there are no NLE baseline models for these datasets to compare with.
>
> Therefore, to address your concern, we ran experiments on the COCO dataset, a dataset differing in distribution from ImageNet (in terms of context and object composition in images). We used zero-shot image captioning as the task. It is worth noting that the ground-truth annotated captions are not considered as explanations, because they are descriptions of images written by humans. However, they serve as a good approximations for explanations. We report results on the common Karpathy test split, and compare our method with two other current state-of-the-art zero-shot baselines in the literature: ZeroCap and ConZic. We use the common metrics for evaluation: BLEU-4 (B4), METEOR (M), CiDEr (C) and SPICE (S). During the past few days, we were able to conduct experiments for 6 models. The results are shown below and they show that our method with BeiT-L, outperforms the baselines on the most important metrics in the captioning task (SPICE , CiDEr) and sets a new state-of-the-art on zero-shot image captioning.
>
> | Method|B4|M|C|S|
> |-|:-:|:-:|:-:|:-:|
> |ZeroCap|**2.6**|11.5|14.6| 5.5|
> |ConZIC|1.3|11.5|12.8|5.2|
> | **Ours**|||||
> |ResNet50|1.7|11.2|14.9|6.6|
> |ResNet101|1.8|11.3|15.5|6.6|
> |DINOv2-B|1.8|11.6|16.2|7.1|
> |ConvNeXtV2-B (pt-384)|2.0|11.5|16.9|7.2|
> |ViT-B/16 (pt)|1.7|**11.9**|17.4|7.4|
> |BEiT-L/16|1.8|11.7|**17.6**|**7.6**|
>
> > Did the authors ablate the prompt template? CLIP can be sensitive to prompt templates, was a similar trend seen for the learnt MLP?
>
> To investigate this matter, we took our pretrained MLP for ResNet50 (trained using the prompt: “an image of a {class}”), and then evaluated it on the Generalized Zero-Shot accuracy for the unseen test classes, using several prompts at the text encoder. We order the prompts by increasing order of sensitivity.
>
> |Prompt|Test@1|Test@5|
> |:-|-:|-:|
> |an image of a {}.|33.28|66.20|
> |an image of the {}.|32.92|65.02|
> |an image of one {}.|31.98|64.06|
> |an image of a large {}.|31.52|63.86|
> |an image of a nice {}.|31.18|64.86|
> |an image of a weird {}.|30.90|64.24|
> |a cropped image of a {}.|30.22|63.42|
> |a black and white image of the {}.|30.14|61.74|
>
> As we can see, similar to CLIP models, the MLP is also sensitive to prompts. Adding or changing one word can result in degradation of zero-shot accuracy. However, this problem can be alleviated by averaging the prompts (as what CLIP does).
>
> > Were there any biases in the generated explanations?
>
> We discuss in detail the biases and ethical concerns in the Limitations section. We kindly refer you to section A in the appendix (page 15).
>
> > Based on Table 1, the simple method seems to work less well for bigger models such as ViT in the generalized zero-shot setting. A little discussion about Table 1 would be helpful.
>
> We analyzed all our models across their original performance, their performance on zero-shot accuracy with our method, and their weight embedding size. We find the following: (i) better-performing models have more discriminative features that are more distributed across space than less-performing models, making it more difficult to regress the weights $W$; (ii) ViT have typically smaller dimensions of $W$ (e.g., 768 or 1024) which leads to a smaller number of constraints when we are learning the regression problem to find $W$; (iii) The $W$ of ViTs have a lack of inductive bias and local structure as opposed to CNNs leading to more difficulty in regressing them.

---

> > ### Comment · Reviewer_zJsF · 2024-11-19
> >
> > Thanks for addressing my comments. I would request you to add the new experiments as part of the paper as they are insightful. Furthermore, I would also recommend the authors to discuss the limitations in more details and add the COCO experiments in the paper or supplementary material.
> >
> > I wish to stay with my score.

---

### Meta-Review · Area_Chair_BVya · 2024-12-20

**Metareview:**

The paper proposes an approach for Natural Language Explanations (NLEs) for vision models. The paper received positive reviews from all 4 reviewers. The reviewers like the new perspective on faithfulness, the motivation behind this work, the simplicity and soundness of the proposed approach, and the comprehensiveness of the experiments. Thus, I recommend acceptance. The authors are strongly encouraged to incorporate reviewers' suggestions, especially on expanding the discussion of limitations (fine-grained/OOD classes) and design decisions of the proposed model.

**Additional Comments On Reviewer Discussion:**

The main concerns revolve around the presentation, justification of design components, and discussion of limitations. Reviewer 9tNT and Reviewer Y6Z6 indicated they are satisfied with the authors' responses. The other two kept their scores after seeing additional experimental results and responses.

---

### Decision · Program_Chairs · 2025-01-22

Accept (Poster)